# SeRL: Self-Play Reinforcement Learning for Large Language Models with Limited Data

**Wenkai Fang**[1], **Shunyu Liu**[2]✉, **Yang Zhou**[1], **Kongcheng Zhang**[1],
**Tongya Zheng**[3], **Kaixuan Chen**[1, 4], **Mingli Song**[1, 4], **Dacheng Tao**[2]

[1]Zhejiang University
[2]College of Computing and Data Science, Nanyang Technological University, Singapore
[3]Hangzhou City University
[4]Hangzhou High-Tech Zone (Binjiang) Institute of Blockchain and Data Security
`wenkfang@zju.edu.cn, shunyu.liu@ntu.edu.sg, imzhouyang@zju.edu.cn,`
`zhangkc@zju.edu.cn, doujiang_zheng@163.com, chenkx@zju.edu.cn,`
`brooksong@zju.edu.cn, dacheng.tao@gmail.com`

## Abstract

Recent advances have demonstrated the effectiveness of Reinforcement Learning (RL) in improving the reasoning capabilities of Large Language Models (LLMs). However, existing works inevitably rely on high-quality instructions and verifiable rewards for effective training, both of which are often difficult to obtain in specialized domains. In this paper, we propose *Self-play Reinforcement Learning* (SeRL) to *bootstrap* LLM training with limited initial data. Specifically, SeRL comprises two complementary modules: self-instruction and self-rewarding. The former module generates additional instructions based on the available data at each training step, employing robust online filtering strategies to ensure instruction quality, diversity, and difficulty. The latter module introduces a simple yet effective majority-voting mechanism to estimate response rewards for additional instructions, eliminating the need for external annotations. Finally, SeRL performs conventional RL based on the generated data, facilitating iterative self-play learning. Extensive experiments on various reasoning benchmarks and across different LLM backbones demonstrate that the proposed SeRL yields results superior to its counterparts and achieves performance on par with those obtained by high-quality data with verifiable rewards. Our code is available at `https://github.com/wantbook-book/SeRL`.

## 1  Introduction

Recent breakthroughs of Large Language Models (LLMs) have demonstrated their potential to revolutionize a wide range of fields [40, 90, 91], such as healthcare [4, 31, 51], robotics [25, 37, 61], and web services [16, 41, 67, 72]. Notably, frontier models such as OpenAI-o3 [49] and DeepSeek-R1 [6] have proven that Reinforcement Learning with Verifiable Rewards (RLVR) can endow LLMs with advanced reasoning capabilities [16, 39, 67, 71, 85, 86], enabling significant achievements in mathematics [6, 76] and programming [21, 26]. A key factor in these developments is the availability of large-scale supervised datasets [7, 14, 78, 80], which incorporate instructions and corresponding ground-truth labels for reward computation.

However, in specialized domains such as clinical diagnostics [44, 58, 60] and aerospace engineering [5, 36, 75], collecting such high-quality data and verifiable supervision remains a significant challenge. This difficulty stems from the fact that acquiring human-labeled data in these areas is particularly labor-

---

✉Corresponding author.

intensive, often requiring substantial expert knowledge and considerable time investment [55, 59]. Moreover, such human-generated annotations are inherently difficult to scale. As an alternative, data generated by LLMs offers a more scalable and cost-effective solution [12, 13, 30, 79, 82]. Nevertheless, this approach usually depends on access to highly capable expert-level LLMs, which may not always be readily available.

To remedy this issue, recent efforts have explored the self-instruction paradigm [11, 27, 63], where LLMs autonomously generate instructions and corresponding responses. While effective for supervised fine-tuning [10, 17, 88], these methods are not directly applicable to online RL settings, which pose two critical challenges. First, existing methods typically generate a fixed dataset without accounting for the evolving capabilities of the LLM during online learning, rendering the static data progressively less suitable. Second, directly treating model-generated responses as ground-truth labels often introduces noise or inconsistency, leading to unreliable reward signals for RL.

In this work, we propose *Self-play **R**einforcement **L**earning*, termed as SeRL, where LLMs *bootstrap* training via autonomous instruction generation and reward estimation under limited initial data. Technically, SeRL consists of two key modules: (1) The self-instruction module iteratively performs few-shot generation to obtain new instructions based on the initial data, producing instructions at each training step that align with the current capability of the model. We introduce a robust online filter that removes low-quality or redundant instructions and enforces difficulty suited to the current ability of the LLM. (2) The self-rewarding module employs majority voting over sampled responses, assigning high rewards to those aligning with the consensus. This enables effective reward estimation without relying on verifiable labels. Based on these two modules, we perform unsupervised RL training on the generated data. Our key contributions are summarized as follows:

- We explore how to leverage RL to incentivize the reasoning abilities of LLMs in data-scarce scenarios, a highly practical yet underexplored challenge for the LLM research community.

- We propose the SeRL framework, comprising two core modules: self-instruction, which leverages few-shot generation to obtain high-quality instructions from limited data, and self-rewarding, which estimates rewards through majority voting without relying on external supervision.

- Extensive experiments on diverse benchmarks demonstrate that the proposed SeRL, even with limited initial data, outperforms other advanced self-play counterparts and achieves results comparable to the baseline trained on full high-quality data with verifiable rewards.

## 2 Background

**Reinforcement Learning for LLMs.** We denote the LLM parameterized by $\theta$ as $\pi_\theta$. For language generation, given an instruction $\boldsymbol{x}$, the model outputs a sequence $\boldsymbol{y} = (y_1, \ldots, y_T)$, where $y_t \sim \pi_\theta(\boldsymbol{x}, \boldsymbol{y}_{<t})$ and $\boldsymbol{y}_{<t} = (y_0, \ldots, y_{t-1})$. RL [22, 38, 56] has recently proven effective in improving the logical reasoning abilities of LLMs. In the RL setting, we additionally require a reward signal. We denote the reward for a given input-output pair $(\boldsymbol{x}, \boldsymbol{y})$ as $R(\boldsymbol{x}, \boldsymbol{y})$. In this work, while our framework is not tied to any specific RL algorithm, we choose Reinforce++ [20] for its robustness and stable performance across diverse tasks. The RL training objective is defined as follows:

$$\mathcal{J}_{\text{RL}}(\theta) = \mathbb{E}_{\boldsymbol{x} \sim \mathcal{D}, \boldsymbol{y} \sim \pi_{\theta_{\text{old}}}} \left[ \frac{1}{|y|} \sum_{t=1}^{|y|} \min \left( \frac{\pi_\theta(y_t|\boldsymbol{x}, \boldsymbol{y}_{<t})}{\pi_{\theta_{\text{old}}}(y_t|\boldsymbol{x}, \boldsymbol{y}_{<t})} \hat{A}(s_t, y_t), \right. \right.$$
$$\left. \left. \text{clip} \left( \frac{\pi_\theta(y_t|\boldsymbol{x}, \boldsymbol{y}_{<t})}{\pi_{\theta_{\text{old}}}(y_t|\boldsymbol{x}, \boldsymbol{y}_{<t})}, 1 - \epsilon, 1 + \epsilon \right) \hat{A}(s_t, y_t) \right) \right], \quad (1)$$

where $\mathcal{D}$ is the dataset of instructions, $s_t = (\boldsymbol{x}, \boldsymbol{y}_{<t})$, $\epsilon$ is a clipping hyperparameter introduced in PPO [53] to stabilize training, and $\pi_{\theta_{\text{old}}}$ denotes the model parameters before the most recent update. Different RL algorithms compute the advantage term $\hat{A}(s_t, y_t)$ in different ways. The details of the Reinforce++ algorithm are provided in Appendix C.1.

**Self-Instruction.** We denote the initial dataset as $\mathcal{D}_{\text{seed}} = \{\boldsymbol{x}_i\}_{i=1}^N$, where $N$ is small in data-scarce domains and insufficient for effective RL training. To address this, we generate a new batch of instructions with a rollout batch size $n_{\text{rbs}}$ at each training step $t$. The generated batch at step $t$ is denoted as $\mathcal{B}_{\text{gen}}^t = \{\boldsymbol{x}_i\}_{i=1}^{n_{\text{rbs}}}$. This step-wise generation better matches the evolving capabilities of

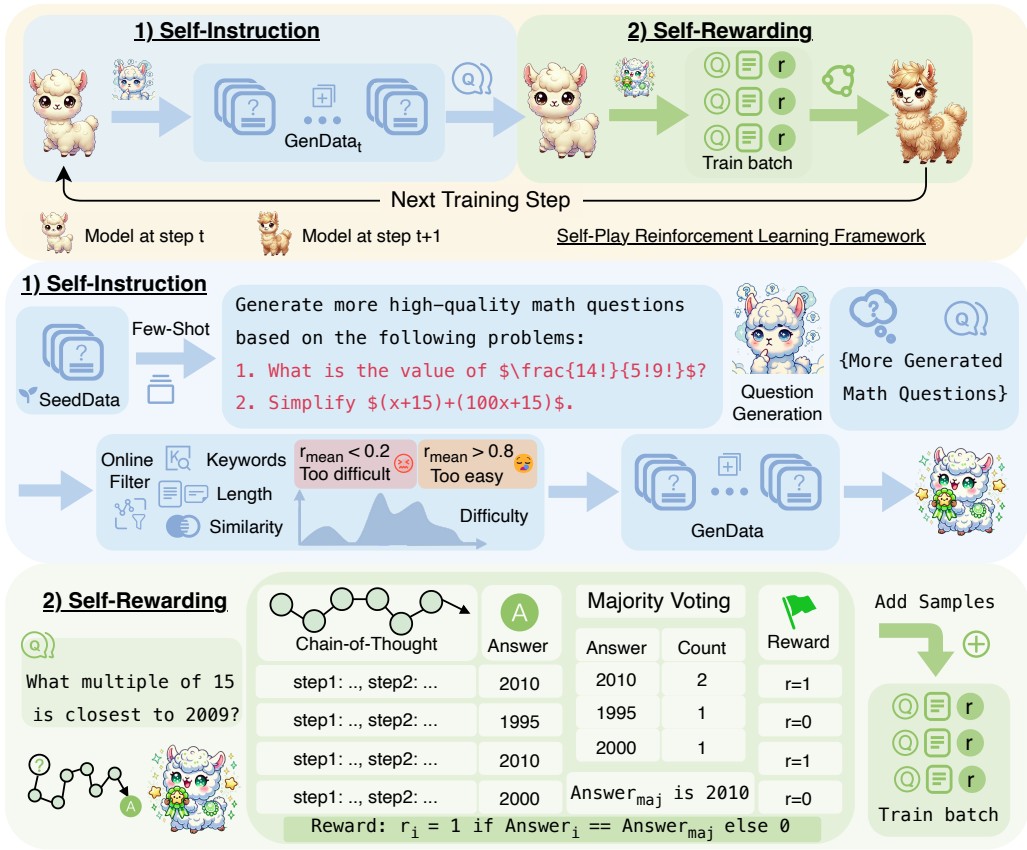

Figure 1: An overview of the proposed SeRL framework, which comprises two core components: (1) Self-Instruction, where the model generates new instructions from a small initial dataset and applies a robust online filtering strategy to ensure instruction quality, diversity, and appropriate difficulty. (2) Self-Rewarding, where the model performs unsupervised RL training using a majority-voting reward mechanism without relying on verifiable labels.

the model. We define the full generated dataset for iteration $n$ as $\mathcal{D}_{\text{gen}}^n = \bigcup_{t \in \mathcal{T}^n} \mathcal{B}_{\text{gen}}^t$, where $\mathcal{T}^n$ represents the set of all generation steps within iteration $n$.

## 3 Self-play Reinforcement Learning

In this work, we introduce SeRL to tackle tasks in data-scarce scenarios. SeRL addresses the challenge by expanding the limited initial dataset and enabling unsupervised reinforcement learning without relying on verifiable labels. SeRL comprises two core components: (1) Self-Instruction. Given a small set of initial data, the model generates additional instructions using few-shot generation. To ensure high data quality, sufficient diversity, and appropriate difficulty, we employ a robust online filtering strategy. (2) Self-Rewarding. We design a self-rewarding mechanism based on majority voting, which allows the model to estimate rewards accurately without the need for verifiable labels, thus supporting fully unsupervised RL training.

### 3.1 Self-Instruction

In data-scarce scenarios, we aim to generate additional instructions from the limited available data to support RL training. To this end, we design the self-instruction mechanism, as illustrated in the blue region of Fig. 1, which consists of the following two steps.

**Instruction Generation.** The LLM is prompted in a few-shot manner to generate new instructions. Each few-shot prompt consists of randomly selected examples from the initial dataset and examples

from the already generated dataset. The reason why we use instructions from the initial dataset is that we aim to expand the dataset while maintaining the same distribution as the initial one, enabling the model to effectively learn the distribution of scarce data. *The reason we do not rely solely on the initial dataset as few-shot examples is that, as training progresses, we aim to adapt the examples to the evolving capabilities of the model.* Specifically, we also selectively incorporate recently generated instructions as few-shot examples for subsequent generation, enabling the model to produce instructions that better match its current level of competence. The prompt template of instruction generation is provided in Tab. C.4.

**Online Instruction Filter.** To ensure the quality, diversity, and moderate difficulty of the generated instructions, we implement a robust online filtering strategy. Specifically, we discard instructions that meet any of the following criteria: (1) a ROUGE-L [35] score exceeding a predefined threshold with existing instructions, to prevent generating semantically similar instructions that reduce diversity; (2) the presence of specific keywords such as "image", "graph", or "picture", which refer to visual content that LLMs cannot process; (3) excessively long or short, since long instructions often contain repetitive phrasing while short ones tend to lack necessary context, both leading to invalid instructions; (4) having a majority answer ratio outside a specified range, ensuring that the retained instructions maintains suitable difficulty for training. Formally, we retain only instructions whose average reward $r_{\text{mean}}$ falls within two tunable bounds, $\gamma_{\text{diff}} \leq r_{\text{mean}} \leq \gamma_{\text{easy}}$. $r_{\text{mean}}$ is calculated as the majority answer ratio from multiple responses of the instruction in our majority-voting reward method.

*The motivation behind this dual-end clipped difficulty filtering strategy is to avoid scenarios where the responses are either entirely correct without any information gain or entirely incorrect with no feasibility for improvement.* Such extreme cases result in zero advantage in algorithms like Reinforce++ [20] and GRPO [54], which can lead to gradient vanishing [65, 42] and consequently significant model degradation. In the ablation studies presented in Section 4.3, we demonstrate that removing the difficulty filtering leads to reward hacking.

## 3.2 Self-Rewarding

In our self-instruction setup, only instructions are generated, and the absence of verifiable labels presents a key challenge for reward computation. Interestingly, Yue et al. [83] suggests that RL with verifiable labels can be viewed as converting the Pass@K performance of a model into Pass@1. Inspired by this insight, we explore whether a similar benefit can be obtained in the absence of verifiable labels. To achieve this, as illustrated in the green region of Fig. 1, *we introduce a majority-voting self-rewarding that serves two purposes:* (1) it improves inference efficiency by enabling the model to reach Maj@N performance with just a single response (Pass@1) after RL training, (2) and it offers a straightforward yet effective way to estimate rewards without the need for verifiable labels. For a given instruction $x_i$, we sample $n_{\text{vote}}$ responses, each consisting of a chain-of-thought reasoning $c_i^k$ and a final answer $a_i^k$:

$$\left(c_i^k, a_i^k\right) \sim \pi_\theta(x_i), \quad \forall x_i \in \mathcal{D}, \ k \in \{1, 2, \ldots, n_{vote}\}. \tag{2}$$

Since we adopt a majority-voting reward estimation method, we define the majority answer for a given instruction $x_i$ as $\text{Maj}\left(\{a_i^k\}_{k=1}^{n_{\text{vote}}}\right)$, where $\{a_i^k\}_{k=1}^{n_{\text{vote}}}$ denotes the set of $n_{\text{vote}}$ answers sampled for $x_i$. For each $a_i^k$, we compute the corresponding reward $R_i^k = \text{Verify}\left(a_i^k, \text{Maj}\left(\{a_i^k\}_{k=1}^{n_{\text{vote}}}\right)\right)$. The Verify function is defined as:

$$\text{Verify}\left(a_i^k, \text{Maj}\left(\{a_i^k\}_{k=1}^{n_{\text{vote}}}\right)\right) = \begin{cases} 1 & \text{if } a_i^k = \text{Maj}\left(\{a_i^k\}_{k=1}^{n_{\text{vote}}}\right), \\ 0 & \text{otherwise.} \end{cases} \tag{3}$$

As instances with all-correct or all-incorrect responses have already been eliminated by the online filter, a majority answer can always be identified. If there is a tie among multiple majority answers, the one with the shortest length is selected based on the principle that a more concise response is preferred to reduce verbosity. We refer to this self-rewarding strategy as majority-voting reward. It serves as a replacement for traditional reward computation methods that rely on reward models or verifiable labels, providing a practical and efficient approach to reward estimation.

## 3.3 Overall Framework

To enable continual improvement in data-scarce scenarios, we adopt a combination of self-instruction and self-rewarding. As shown in the yellow region of Fig. 1, the model iteratively generates new

Table 1: Pass@1 comparison across benchmarks between our method and baseline approaches, with all models evaluated using greedy decoding. Initial refers to the original LLaMA-3.2-3B-Instruct or Qwen-2.5-7B-Instruct model. **Bold** indicates the best performance.

| Models | Iteration | Methods | MATH-500 | MATH-Hard | ASDiv | College Math | TabMWP |
|---|---|---|---|---|---|---|---|
| LLaMA-3.2 -3B-Instruct | | Initial | 47.6 | 22.5 | 84.6 | 35.2 | 46.4 |
| | | RL-GT | 49.7 | **24.1** | 88.9 | 36.4 | 69.9 |
| | Iter1 | SR-DPO | 42.4 | 20.4 | 80.9 | 31.5 | 42.0 |
| | | I-RPO | 44.0 | 17.8 | 86.1 | 32.2 | 58.9 |
| | | SeRL | 48.6 | 23.0 | 87.5 | 36.7 | 68.4 |
| | Iter2 | SR-DPO | 38.3 | 19.4 | 61.8 | 27.9 | 34.1 |
| | | I-RPO | 42.9 | 18.6 | 87.8 | 31.1 | 58.2 |
| | | SeRL | 50.4 | 23.6 | 88.9 | **38.2** | **72.3** |
| | Iter3 | SR-DPO | 32.7 | 17.6 | 57.5 | 23.6 | 29.5 |
| | | I-RPO | 42.5 | 18.1 | 87.6 | 27.9 | 52.2 |
| | | SeRL | **52.6** | 23.7 | **89.0** | 37.7 | 70.6 |
| Qwen-2.5 -7B-Instruct | | Initial | 74.2 | 48.8 | 93.5 | 54.3 | 75.5 |
| | | RL-GT | 74.8 | **50.9** | 94.3 | **55.5** | 72.7 |
| | Iter1 | SR-DPO | 72.6 | 49.2 | 94.0 | 54.6 | **80.7** |
| | | I-RPO | 71.4 | 47.6 | 91.4 | 53.8 | 71.6 |
| | | SeRL | 74.2 | 50.0 | 94.2 | 54.3 | 77.0 |
| | Iter2 | SR-DPO | 73.8 | 50.0 | 94.1 | 55.0 | 78.7 |
| | | I-RPO | 74.0 | 45.2 | 93.9 | 53.2 | 73.7 |
| | | SeRL | 74.8 | 49.7 | **94.7** | 55.1 | 80.1 |
| | Iter3 | SR-DPO | 71.6 | 47.4 | 92.4 | 53.0 | 72.7 |
| | | I-RPO | 72.8 | 46.3 | 92.5 | 52.0 | 71.4 |
| | | SeRL | **75.8** | 50.4 | 94.4 | 55.1 | 79.4 |

instructions from limited data and estimates rewards for sampled responses using majority voting. This process enables sustained reinforcement learning across training iterations. To match the evolving capability of the model, we generate a batch of instructions at each training step $t$ using self-instruction. For each instruction in $\mathcal{B}^t_{\text{gen}}$, we sample $n_{vote}$ responses and compute the majority-voting reward. The resulting triplets $\{(\boldsymbol{x}_i, \boldsymbol{y}_i, r_i)\}$, consisting of the instruction, sampled response, and computed reward, are used to update the model parameters in one training step. At the next training step $t+1$, we repeat the same process to generate $\mathcal{B}^{t+1}_{\text{gen}} = \{\boldsymbol{x}_i\}^{n_{\text{rbs}}}$, followed by one step of RL training. This cycle repeats across iterations, gradually refining the performance of the model.

## 4 Experiments

### 4.1 Experimental Setup

**Models and Training Settings.** We conducted experiments using two model series: LLaMA-3.2-3B-Instruct [46] and Qwen-2.5-7B-Instruct [78]. We simulate a data-scarce scenario in the mathematics domain by limiting the initial dataset to 500 instructions, which are uniformly sampled across all difficulty levels from the MATH training set [18]. We denote this initial set of 500 instructions as $\mathcal{D}_{\text{seed}}$. For training, we adopt the OpenRLHF [19] framework and employ its Reinforce++ [20] algorithm for RL. Detailed training hyperparameters are provided in Tab. 6 and Tab. 7.

**Baselines.** We compare our method against two multi-round iterative training baselines and one RL baseline with ground-truth rewards (RL-GT): (1) Iterative RPO (I-RPO) [50], which runs for three iterations. In each iteration, four responses are sampled for each instruction in the MATH training set. Each response is scored using a rule-based reward, and the highest and lowest are selected as the chosen and rejected responses for DPO training. Rule-based methods [54, 50] determine correctness

Table 2: Multi-round performance of our proposed SeRL on MMLU-Pro.

| Methods | LLaMA-3.2-3B-Instruct | | | | | Qwen-2.5-7B-Instruct | | | | |
| | STEM | Humanities | Social | Other | Avg. | STEM | Humanities | Social | Other | Avg. |
|---|---|---|---|---|---|---|---|---|---|---|
| Initial | 32.1 | 26.7 | 41.9 | 34.3 | 34.0 | 59.9 | 40.1 | 64.1 | 53.9 | 57.3 |
| RL-GT | 34.6 | 27.6 | 44.1 | 36.0 | 36.1 | 60.0 | 39.6 | 63.9 | 53.1 | 57.2 |
| SeRL (iter1) | 32.6 | 27.8 | 41.5 | 34.0 | 34.3 | 60.0 | 40.8 | 64.2 | 54.3 | 57.5 |
| SeRL (iter2) | 33.8 | 27.7 | 41.9 | 33.9 | 35.0 | 60.2 | 40.0 | 63.9 | 53.5 | 57.3 |
| SeRL (iter3) | 33.9 | 28.1 | 41.9 | 34.1 | 35.1 | 59.8 | 40.9 | 63.9 | 54.1 | 57.4 |

by extracting the answer from the response using regular expressions and comparing it against the ground-truth answer to assign a reward. (2) Self-Rewarding DPO (SR-DPO) [81], also run for three iterations. In each iteration, a new dataset $\mathcal{D}_{\text{gen}} = \{x_i\}_{i=1}^N$ is synthesized based on $\mathcal{D}_{\text{seed}}$ using the same self-instruction approach as ours. For each generated instruction, four responses are sampled, and model-based rewards are used to construct preference pairs for DPO training. Model-based methods [81] rely on manually designed principles or scoring rubrics, which the model uses to evaluate the response and assign a reward. (3) RL-GT is run for a single iteration, using the MATH training set for RL training. The rewards are computed using a rule-based reward function.

Since the original reward prompts in SR-DPO are intended for general tasks, we modify them to better fit mathematical scenarios, ensuring a fair comparison (see Tab. C.5). To ensure fairness in training data scale, we note that RL-GT and I-RPO use 7,500 instructions from the MATH training set, and SR-DPO generates 7,500 instructions per iteration. For our method, we online-generate instructions during training, keeping the same number of training gradient steps as other baselines to ensure the total training data scale remains consistent, thereby guaranteeing a fair comparison.

**Evaluation Benchmarks.** We evaluate model performance using five math-specific benchmarks and one general-purpose benchmark. (1) The math benchmarks include MATH-500 [34], which focuses on medium-difficulty problems; MATH-Hard [18], which contains level-5 competition problems from AMC and AIME to assess advanced reasoning; ASDiv [47], a dataset of 2,305 diverse elementary-level math word problems; College Math [57], which covers university-level topics such as algebra and calculus; and TabMWP [43], consisting of 38,431 problems that combine textual and tabular data to test multi-step reasoning. (2) To assess general reasoning ability, we include MMLU-Pro [64], an enhanced version of MMLU covering STEM, humanities, and social sciences. All evaluations are conducted using greedy decoding with a maximum of 1,024 new tokens.

## 4.2 Experimental Results

**Our approach matches the performance of the method using extensive data with verifiable rewards.** As shown in Tab. 1, after just the first round of unsupervised RL training on generated data, our model achieves performance comparable to RL-GT. After the second round, LLaMA-3.2-3B-Instruct even outperforms RL-GT across nearly all evaluation benchmarks. LLaMA-3.2-3B-Instruct continues to improve in the third round. After three rounds of iterative training, Qwen-2.5-7B-Instruct also outperforms RL-GT on the MATH-500, ASDiv, and TabMWP benchmarks. Additional results of our method for more iterations are provided in Appendix D.2.

**Our method outperforms other multi-round iterative approaches.** Across all iterations, our method consistently outperforms both SR-DPO and I-RPO on LLaMA-3.2-3B-Instruct. We observe that both SR-DPO and I-RPO exhibit noticeable model degradation over multiple iterations. For SR-DPO, it performs poorly on LLaMA-3.2-3B-Instruct, while showing slightly better results on Qwen-2.5-7B-Instruct. This suggests that model-based reward mechanisms may depend on model scale, as larger models are more likely to produce accurate reward estimates. To better understand the limited effectiveness of SR-DPO, we further analyze its reward estimation accuracy in Section 4.3 (c). In contrast, I-RPO shows a decline in performance over multiple training rounds on both models. We hypothesize that this degradation results from the inclusion of a negative log-likelihood (NLL) loss term in its training objective, which may cause the model to overfit the chosen responses. As a consequence, this overoptimization reduces generalization and ultimately leads to performance drops.

**Our method generalizes well to general reasoning tasks.** To better evaluate generalization, we assess our method on the MMLU-Pro benchmark. As shown in Tab. 2, LLaMA-3.2-3B-Instruct achieves consistent gains over multiple iterations in certain categories, whereas Qwen-2.5-7B-Instruct

Table 3: Pass@1 comparison under the same number of training steps with different filtering strategies ablated, using LLaMA-3.2-3B-Instruct.

| Methods | MATH-500 | MATH-Hard | ASDiv | College Math | TabMWP |
|---|---|---|---|---|---|
| SeRL | 52.6 | 23.7 | 89.0 | 37.7 | 70.6 |
| SeRL w/o Length Filter | 47.6 | 23.2 | 87.4 | 36.2 | 60.1 |
| SeRL w/o Keywords Filter | 48.0 | 22.1 | 87.4 | 36.3 | 62.5 |
| SeRL w/o Similarity Filter | 48.8 | 23.3 | 87.3 | 35.6 | 64.6 |
| SeRL w/o Difficulty Filter | 11.6 | 5.1 | 1.5 | 14.0 | 10.2 |

shows minimal improvement. We attribute this to Qwen-2.5 already being extensively fine-tuned on MATH data, as further rule-based RL training on the MATH training set also fails to improve its MMLU-Pro performance. LLaMA-3.2-3B-Instruct shows consistent improvement in the STEM category, with additional gains in Humanities, eventually surpassing RL-GT. Performance in the Social and Other categories remain largely unchanged. This is likely because our training data is math-focused, enhancing reasoning skills that benefit STEM tasks. The improvement in Humanities is mainly due to the Law subcategory, where many questions involve logical reasoning. As the reasoning ability of the model improves, its performance on Law tasks also increases.

## 4.3 Ablation Study

**(a) The dual-end clipped difficulty filtering mechanism effectively prevents reward hacking.** As shown in Fig. 2, when we remove the difficulty filtering, the model exhibits clear signs of reward hacking: the average reward continues to increase with training steps, but the accuracy on the MATH-500 test set drops sharply after around 100 steps.

To investigate this issue, we perform a case study (Tab. E.2) and observe that although the intermediate reasoning is often correct, the model consistently ends with the final statement: "The final answer is: $\boxed{0}$." This behavior arises because our reward function is based on majority voting. If all $n_{vote}$ sampled responses from the LLM produce the same (albeit incorrect) answer, such as $\boxed{0}$, each response receives a high reward, despite being a wrong answer. The root cause of this phenomenon lies in the difficulty of the instruction. When an instruction is too hard, the model struggles to produce consistent answers across samples, leading to unstable or misleading reward signals. To address this, we introduce a dual-end dif-

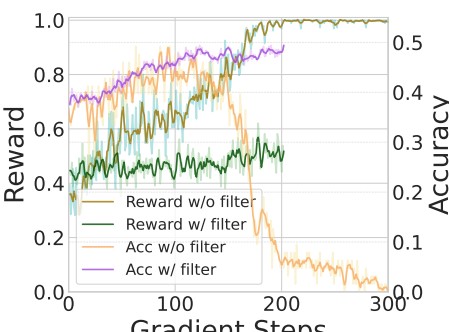

Figure 2: Training curve of LLaMA-3.2-3B-Instruct with and without the online filter.

ficulty filtering mechanism to filter out instructions with insufficient answer agreement, i.e., those with $r_{\text{mean}} < \gamma_{\text{diff}}$. Similarly, overly easy instructions do not contribute meaningfully to learning and are also filtered by applying the upper threshold $r_{\text{mean}} > \gamma_{\text{easy}}$. With this dual-end clipped difficulty filtering mechanism, the reward hacking issue is mitigated in subsequent experiments.

**(b) All of our proposed data filtering strategies demonstrate their effectiveness.** To demonstrate the effectiveness of each filtering strategy in the online filtering process, we conducted an ablation study as shown in Tab. 3. The results show that every filtering strategy contributes to the model's performance, as removing any of them leads to a decline in overall results. In particular, omitting the difficulty filter may cause reward hacking, as illustrated in Fig. 2.

**(c) Our method effectively mitigates the bias inherently introduced by self-instruction and self-rewarding.** Our proposed method is specifically designed to minimize this bias as much as possible, and matches the performance of methods using extensive data with verifiable rewards, despite the limited amount of seed data used in our method.

For the self-instruction module, we introduce an online filtering strategy that maintains data quality while promoting diversity in the generated samples. We have provided a comprehensive analysis

Table 4: Comparison of training results between RL-GT and RL-MV (Ours).

| Models | Methods | MATH-500 | MATH-Hard | ASDiv | College Math | TabMWP |
|---|---|---|---|---|---|---|
| LLaMA3.2-3B-Instruct | RL-GT | 49.7 | 24.1 | 88.9 | 36.4 | 69.9 |
| | RL-MV (Ours) | 49.8 | 24.8 | 88.6 | 37.2 | 72.4 |
| Qwen2.5-7B-Instruct | RL-GT | 74.8 | 50.9 | 94.3 | 55.5 | 72.7 |
| | RL-MV (Ours) | 75.4 | 51.2 | 94.4 | 55.5 | 74.5 |

of the generated data in terms of quality, difficulty, and diversity, as detailed in Appendix F. These results demonstrate the reliability of our self-instruction method for data generation.

For the self-rewarding module, as stated in the related works [81, 87, 92], estimation bias is inevitable in self-rewarding settings. Our majority-voting reward method assigns rewards based on the consistency among multiple sampled responses, offering greater stability than scoring individual responses. To demonstrate this, we calculate the cosine similarity between the estimated rewards and the ground-truth rewards, with higher similarity indicating greater accuracy in reward estimation. As shown in Fig. 3, on both LLaMA-3.2-3B-Instruct and Qwen-2.5-7B-Instruct, the majority-voting reward demonstrates significantly higher alignment with the rule-based reward, indicating superior accuracy in reward estimation. In contrast, the low accuracy of the model-based reward on LLaMA-3.2-3B-Instruct likely accounts for the underperformance of SR-DPO on this model. Additionally, the model-based reward shows higher consistency with the rule-based reward on Qwen-2.5-7B-Instruct, which accounts for the relatively better performance of SR-DPO on Qwen-2.5-7B-Instruct. More comparisons of reward methods and alignment metric results can be found in Appendix E.1.

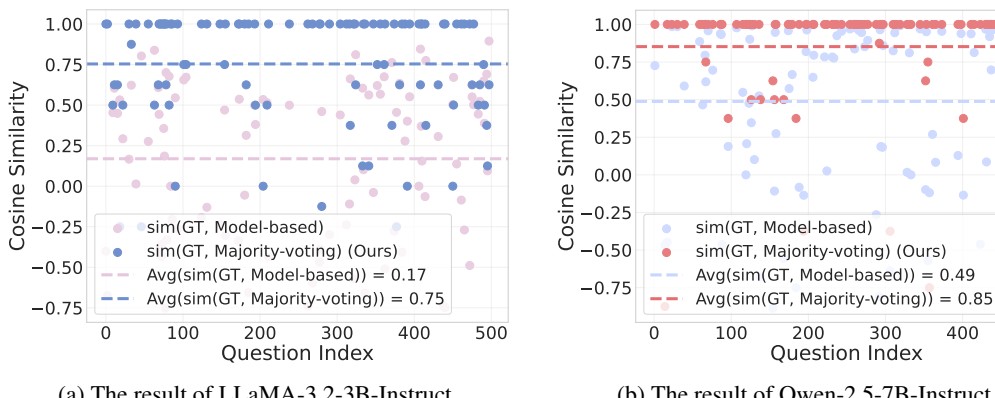

(a) The result of LLaMA-3.2-3B-Instruct.  (b) The result of Qwen-2.5-7B-Instruct.

Figure 3: Cosine similarity between rule-based reward and majority-voting / model-based reward on MATH-500. Each point represents the similarity over 16 sampled responses per instruction. "sim(GT, Model-based)" refers to the cosine similarity between the rule-based reward and the model-based reward, while "sim(GT, Majority-voting)" refers to the cosine similarity between the rule-based reward and our majority-voting reward. The dashed lines in the figure indicate the average values.

To further assess the effectiveness of our majority-voting reward, we conduct an ablation study shown in Tab. 4. In this setting, we remove the self-instruction mechanism and perform one round of majority-voting reward RL using the full MATH training set, denoted as RL-MV. Results show that this variant achieves performance comparable to RL-GT across all test datasets. Notably, LLaMA-3.2-3B-Instruct outperforms RL-GT on MATH-500, MATH-Hard, College Math, and TabMWP, while Qwen-2.5-7B-Instruct consistently surpasses RL-GT on nearly all benchmarks.

**(d) Our method is applicable to different RL algorithms.** To verify the robustness of our method across different RL algorithms, we evaluate it under various RL setups. RLOO [1] and Reinforce++ [20] use $k_1$ KL divergence estimation, while GRPO [54] adopts $k_3$ KL divergence estimation. The formulations for both $k_1$ and $k_3$ KL divergence estimations are detailed in Appendix B. Since $k_1$ is integrated into the per-token reward term, we set a smaller KL coefficient for training RLOO and

Table 5: Comparison of SeRL performance under different reinforcement learning algorithms. **Bold** indicates the best performance.

| Models | Algorithm | Methods | MATH-500 | MATH-Hard | ASDiv | College Math | TabMWP |
|--------|-----------|---------|----------|-----------|-------|--------------|--------|
| LLaMA-3.2-3B-Instruct | RLOO | SeRL(iter1) | 49.1 | 24.3 | 87.7 | 36.0 | 62.7 |
| | | SeRL(iter2) | 49.2 | **24.7** | 88.9 | 37.7 | 70.1 |
| | | SeRL(iter3) | 51.7 | 23.8 | **89.0** | 36.7 | 70.3 |
| | GRPO | SeRL(iter1) | 50.4 | 23.4 | 88.0 | 36.6 | 64.8 |
| | | SeRL(iter2) | 49.2 | 23.2 | 88.4 | 36.0 | 67.3 |
| | | SeRL(iter3) | 48.6 | 24.0 | 88.3 | 36.1 | 67.6 |
| | Reinforce++ | SeRL(iter1) | 48.6 | 23.0 | 87.5 | 36.7 | 68.4 |
| | | SeRL(iter2) | 50.4 | 23.6 | 88.9 | **38.2** | **72.3** |
| | | SeRL(iter3) | **52.6** | 23.7 | **89.0** | 37.7 | 70.6 |

Reinforce++ to avoid strong KL regularization. All other hyperparameter settings remain the same. As shown in Tab. 5, all algorithms achieve comparable performance, demonstrating the generality of our approach. Among them, the Reinforce++ algorithm delivers the best overall results. The GRPO algorithm achieves relatively strong performance in the first iteration but shows continued improvement only on MATH-Hard, ASDiv, and TabMWP in subsequent rounds, while its performance declines on MATH-500 and College Math. Similarly, RLOO exhibits performance degradation on MATH-Hard and College Math after the third iteration.

Reinforce++ demonstrates greater robustness compared to GRPO and RLOO due to the following reasons: (1) GRPO normalizes the estimated advantage over the $n_{\text{vote}}$ responses for each question, whereas Reinforce++ estimates the advantage over all responses across all questions, resulting in a larger group and more stable estimation. In addition, GRPO uses an external $k_3$ KL estimation, which involves an exponential term. This may cause large spikes in the gradient, leading to instability during training. (2) RLOO applies a preprocessing step that subtracts the average reward of other responses from the reward of each individual response, thereby increasing the reward gap between different responses. However, since self-rewarding may introduce inaccurate reward estimations, the amplified reward differences in RLOO could lead to greater bias when the estimation is incorrect. As a result, its performance tends to be worse than that of Reinforce++.

## 5 Related Work

**Self-Instruction Methods.** Recent studies show that RL can significantly enhance the reasoning abilities of LLMs. However, it depends on high-quality instructions and verifiable labels, which are scarce in specialized domains such as clinical diagnostics [58, 44, 60] and aerospace engineering [36, 5, 75]. This challenge has spurred interest in synthetic data generation. Previous work has employed powerful expert models like GPT-4 to generate synthetic data [79, 13, 32, 30, 82], but these methods are costly and rely on external proprietary systems. To mitigate this dependency, self-instruction frameworks such as Wang et al. [63] bootstrap data from a small set of seed examples via few-shot generation and heuristic filtering to ensure quality and diversity. Many other studies [88, 10, 27, 11, 17] have extended the self-instruction framework, focusing on improving data efficiency, diversity, and quality. Some studies train dedicated models for instruction generation. For instance, WizardLM [74] evolves instruction data in terms of complexity, diversity, and quality, while TeaMs-RL [15] extends it by defining prompt-based augmentation actions and training an Instructor LLM via RL to apply them. In contrast, we find that existing models already possess strong instruction-generation abilities. Without training a separate model or designing additional augmentation prompts, our method leverages few-shot generation and an online filtering strategy to produce diverse, high-quality instructions (see Appendix F). Unlike these offline SFT approaches with fixed data, RL enables online data generation that adapts to the model's evolving capability. However, it still requires verifiable labels and mechanisms to prevent reward hacking.

**Self-Rewarding Methods.** Recent successes of RL in LLMs largely rely on large-scale instructions with verifiable rewards [73, 23, 66]. However, labeled data typically requires substantial human effort and is limited in both efficiency and quality. As a more efficient alternative, self-rewarding

methods that estimate rewards without relying on verifiable labels have gained increasing attention. Model-based self-rewarding methods like Bai et al. [2] and Yuan et al. [81] use predefined rules to generate preference pairs for Direct Preference Optimization (DPO) training. Fränken et al. [9] maximize mutual information between principles and responses, while Zhang et al. [87] is based on semantic entropy clustering. RLSC [33] optimizes the model by directly maximizing its most probable response, while our majority-voting method is based on multiple responses, leading to greater stability and more consistent outputs. TTRL [92], as a concurrent work, also employs a majority-voting self-rewarding mechanism in an unsupervised reinforcement learning setting. However, our approach differs in two important aspects: (1) we generate diverse synthetic training instructions from limited data through self-instruction, unlike test-time training of TTRL on fixed test sets; (2) our method supports multiple rounds of iterative updates, enabling continual self-improvement and refinement of the policy, while TTRL only performs a single-round adaptation on a static test set.

## 6    Discussion

**Ceiling of Self-Iteration.** We begin by discussing the performance upper bound of LLMs within a single iteration. When performing RL training using only majority-voting reward, the maximum achievable Pass@1 on the training set is inherently limited by the Maj@K before training. If the LLM has strong generalization capabilities, the Pass@1 upper bound on an in-domain test set should also align with the Maj@N of the model on the test set before training. So, what is the upper bound of LLM capability across multiple iterations? If the Maj@N of the model continues to improve after each round of training, then each iteration defines a new, higher ceiling. In this case, the LLM is capable of approaching increasingly better performance bounds, suggesting the potential for unbounded, continual improvement through iterative self-training. However, as shown in Fig. 7, the Maj@16 performance of both LLaMA-3.2-3B-Instruct and Qwen-2.5-7B-Instruct on MATH-500 does not continue to improve with more iterations. This observation aligns with the analysis by Yue et al. [83], which suggests that RL with verifiable reward primarily transforms Pass@K ability into Pass@1, while the upper bound of a model is constrained by its underlying capability. Nevertheless, our method remains valuable, as it enables efficient reasoning in data-scarce scenarios through unsupervised RL training with data augmentation.

**Limitation of Majority-voting Reward.** In scenarios where there is no deterministic final answer such as writing tasks, or where the correctness of the process is more important such as mathematical proofs, our majority voting strategy may not be applicable. However, for most knowledge-based scenarios, even when a numeric answer is not required as in mathematical problem solving, the question can often be reformulated into a multiple-choice format, making our majority voting approach still applicable. Another limitation is that the reliability of the reward signal in self-rewarding methods can be compromised for challenging problems, as also discussed in prior work [81, 87, 92]. To mitigate this issue, we have designed a difficulty filtering strategy that removes instructions for which the model shows uncertainty in generating consistent answers. This acts as a form of curriculum learning, where the model starts training on simpler problems and gradually moves toward more complex ones. As a result, the estimation bias induced by unreliable self-rewards is reduced during training. While this strategy does not completely resolve the issue, we believe it is a promising direction and worth further exploration in future work.

## 7    Conclusion

In this work, we introduce SeRL, a framework designed to *bootstrap* LLM training from limited initial data. SeRL consists of two key components: a self-instruction module and a self-rewarding module. The self-instruction module expands the initial data by generating new instructions, with robust and effective online filtering applied to ensure quality, diversity, and appropriate difficulty. The self-rewarding module uses a majority-voting strategy to estimate rewards for generated responses, removing the need for external labels. Based on the generated data, SeRL performs standard RL training in an iterative, self-improving manner. Extensive experiments across multiple reasoning benchmarks and LLM architectures show that SeRL consistently outperforms strong baselines and matches the performance of methods trained with large-scale, high-quality labeled data.

## Acknowledgement

This work is supported in part by the Hangzhou Joint Funds of the Zhejiang Provincial Natural Science Foundation of China under Grant No. LHZSD24F020001, in part by the Zhejiang Province High-Level Talents Special Support Program "Leading Talent of Technological Innovation of Ten-Thousands Talents Program" under Grant No. 2022R52046, in part by the Fundamental Research Funds for the Central Universities under Grant No. 2021FZZX001-23, in part by the National Natural Science Foundation of China (62506330), and in part by the advanced computing resources provided by the Supercomputing Center of Hangzhou City University. This research is supported by the RIE2025 Industry Alignment Fund – Industry Collaboration Projects (IAF-ICP) (Award I2301E0026), administered by A*STAR, as well as supported by Alibaba Group and NTU Singapore through Alibaba-NTU Global e-Sustainability CorpLab (ANGEL).

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

# Appendix

**Table of Contents**

## A  Additional Related Works

**Self-Iteration Methods.** Multi-round iterative training enables models to reach their performance upper bound progressively, and self-iteration has been widely explored. Pang et al. [50] constructs preference pairs using rule-based rewards for DPO training. Chen et al. [3] adopts adversarial learning based on human-labeled responses, making the method reliant on label quality. Dong et al. [8] performs DPO-based iteration with a fixed reward model, while Wu et al. [70] formulates training as a zero-sum game using a learned preference model. Rosset et al. [52] proposes DNO, which uses an expert model to estimate win rates by regressing internal rewards through batch-policy iteration. In contrast, we propose a majority-voting reward mechanism that requires no external supervision. Compared to prior offline approaches [81, 84, 69], our method adopts an online RL framework. We also address data-scarce scenarios by generating instructions through self-instruction that adapt to the evolving capabilities of the model, and by using a simple yet effective majority-voting reward for accurate and efficient self-iteration.

## B  KL Divergence Estimators

There exist multiple estimators for KL divergence, such as $k_1$, $k_2$, and $k_3$, defined as follows:

$$k_1(t) = -\log \frac{\pi_\theta(y_t|\boldsymbol{x}, \boldsymbol{y}_{<t})}{\pi_{\text{ref}}(y_t|\boldsymbol{x}, \boldsymbol{y}_{<t})}, \tag{4}$$

$$k_2(t) = \frac{1}{2}\left(\log \frac{\pi_\theta(y_t|\boldsymbol{x}, \boldsymbol{y}_{<t})}{\pi_{\text{ref}}(y_t|\boldsymbol{x}, \boldsymbol{y}_{<t})}\right)^2, \tag{5}$$

$$k_3(t) = -\log \frac{\pi_\theta(y_t|\boldsymbol{x}, \boldsymbol{y}_{<t})}{\pi_{\text{ref}}(y_t|\boldsymbol{x}, \boldsymbol{y}_{<t})} + \exp\left(\log \frac{\pi_\theta(y_t|\boldsymbol{x}, \boldsymbol{y}_{<t})}{\pi_{\text{ref}}(y_t|\boldsymbol{x}, \boldsymbol{y}_{<t})}\right) - 1. \tag{6}$$

GRPO uses the $k_3$ estimator as an external loss component, while Reinforce++ and RLOO adopt the $k_1$ estimator, which is incorporated into the reward as a KL reward.

## C  Implementation Details

All experiments are run on a cluster with 8× NVIDIA RTX A6000 GPUs, a 96-core Intel Xeon Gold 5318Y CPU, and 512 GB RAM.

### C.1  Details of Reinforce++ Algorithm

Assuming a training batch contains $n_{\text{bs}}$ questions and each question is associated with $n_{\text{vote}}$ sampled responses, $G_{ij}(s_t, y_t)$ denotes the return at the $t$-th token of the $j$-th sampled response $\mathbf{y}_{i,j}$ for the $i$-th question $\mathbf{x}_i$. We compute the mean and standard deviation of the return across the batch as:

$$\text{mean}_t = \frac{1}{n_{\text{bs}} \cdot n_{\text{vote}}} \sum_{i=1}^{n_{\text{bs}}} \sum_{j=1}^{n_{\text{vote}}} G_{ij}(s_t, y_t), \ \text{std}_t = \sqrt{\frac{1}{n_{\text{bs}} \cdot n_{\text{vote}}} \sum_{i=1}^{n_{\text{bs}}} \sum_{j=1}^{n_{\text{vote}}} \left(G_{ij}(s_t, y_t) - \text{mean}_t\right)^2}. \tag{7}$$

Then, the advantage is computed as:

$$\hat{A}_{ij}(s_t, y_t) = \frac{G_{ij}(s_t, y_t) - \text{mean}_t}{\text{std}_t}. \tag{8}$$

We would like to further clarify the computation of $G_{ij}(s_t, y_t)$ as follows:

$$G_{ij}(s_t, y_t) = \sum_{k=t}^{|\mathbf{y}_{i,j}|} R_{ij}(s_k, y_k) - KL_{ij}(k), \tag{9}$$

where $R_{ij}(s_k, y_k)$ denotes the token-level reward at the $k$-th token of the $j$-th response to the $i$-th question and $KL_{ij}(k)$ denotes the token-level KL divergence at the $k$-th token between the current

model and the initial model for the $j$-th response to the $i$-th question. A negative sign is applied to serve as a KL penalty. Reinforce++ adopts the $k_1$ estimator.

In practice, the reward is assigned at the response level, consistent with other related works [62, 89, 54]. Therefore, only the last token receives a reward, and the above equation 9 can be simplified as:

$$G_{ij}(s_t, y_t) = R_{ij}(s_T, y_T) - \sum_{k=t}^{T} KL_{ij}(k) = R(\mathbf{x}_i, \mathbf{y}_j) - \sum_{k=t}^{T} KL_{ij}(k). \tag{10}$$

Here, $R(\mathbf{x}_i, \mathbf{y}_j)$ denotes the reward assigned to the $j$-th response of the $i$-th question.

## C.2 Details on Chain of Thought

During the RL sampling stage, we prompt the model with "think step by step and give the final answer", encouraging it to first generate a chain of thought (CoT) and then produce the final answer.

## C.3 Details on Filtering Strategies

The proposed online instruction filter consists of four components:

- (1) The similarity filter computes the ROUGE-L score between each newly generated instruction and all existing instructions in the dataset. If the score exceeds a predefined threshold (set to 0.7 following the prior work [63]), indicating high similarity to existing instructions, the new instruction is filtered out to encourage diversity.

- (2) The keywords filter removes instructions containing specific keywords such as "image", "graph", "picture", "file", "map", "draw", "plot", or "write a program", as they refer to visual content or capabilities beyond the model's scope. In addition, instructions starting with punctuation or non-English characters are also excluded.

- (3) The length filter removes instructions that are either excessively long or short. Instructions exceeding 150 words often contain redundant content or even include solutions, while those with fewer than 3 words typically lack sufficient context for problem solving. This filtering step helps maintain the overall quality and clarity of the generated instructions.

- (4) The difficulty filter evaluates the proportion of majority answers among the generated responses for a given instruction. Instructions with excessively high or low majority proportions are filtered out to ensure that the retained instructions maintains suitable difficulty for model learning. We set the thresholds as $\gamma_{\text{diff}} = 0.2$ and $\gamma_{\text{easy}} = 0.8$, as shown in Tab. 6.

## C.4 Details on Instruction Generation

We adapt the instruction generation prompt from prior self-instruction work [63] to better suit the mathematical domain. The modified prompt template is shown in Box C.1. The $n_{\text{shot}}$ examples used in the few-shot context are randomly sampled from both the seed data and the generated instructions. Specifically, one-fourth ($n_{\text{shot}}/4$) are drawn from the seed data, while the remaining three-fourths ($3n_{\text{shot}}/4$) from previously generated instructions.

## C.5 SR-DPO Rewarding Prompt

Since the prompt used in the original SR-DPO paper [81] is designed for reward estimation on general tasks, it performs poorly on mathematical problems. To ensure a fair comparison in our experiments, we modify the original prompt to better suit the evaluation of mathematical tasks, as shown in Box C.2. Since it is also a self-rewarding method, we use the language model, which is to be trained, as the reward model to estimate the reward.

## C.6 RL Hyperparameters

The experimental settings [19] for training LLaMA-3.2-3B-Instruct and Qwen-2.5-7B-Instruct using our SeRL method are shown in Tab. 6 and Tab. 7.

**Box C.1: Few-shot instruction generation prompt**

```
Please act as a professional math teacher.
Your goal is to create high quality math word problems to help students learn
math.
You only need to create the new question.  Please DO NOT solve it.

Come up with a series of tasks:

Task 1:  {instruction for existing task 1}
Task 2:  {instruction for existing task 2}
Task 3:  {instruction for existing task 3}
Task 4:  {instruction for existing task 4}
Task 5:  {instruction for existing task 5}
Task 6:  {instruction for existing task 6}
Task 7:  {instruction for existing task 7}
Task 8:  {instruction for existing task 8}
Task 9:
```

**Box C.2: Reward estimation prompt in mathematical tasks**

```
Review the user's math question and the corresponding response using the
additive 5-point scoring system described below.  Points are accumulated
based on the satisfaction of each criterion:

- Add 1 point if the response understands the problem and introduces at
least one relevant mathematical concept or formula, even if the derivation
is incomplete or has gaps.

- Add another point if the response provides the main idea or a key result
addressing a substantial part of the problem, yet still contains calculation
mistakes, skipped steps, or loose rigor.

- Award a third point if core computations are broadly correct and the logic
is mostly coherent; only minor slips or omissions remain, and the overall
workflow can be followed and reproduced.

- Grant a fourth point if all calculations are accurate and the reasoning is
rigorous, step-by-step, with each step contributing meaningfully to the final
answer and no irrelevant or redundant content.

- Bestow a fifth point if, in addition to the above, the solution is
expertly presented:  clearly structured and well-formatted, cites necessary
theorems or justifies key steps, proactively verifies the result (e.g.,
substitution, extrema check), and possibly offers alternative insights or
elegant shortcuts-without extra fluff.

User:  <QUESTION>

<response><RESPONSE></response>

After examining the user's instruction and the response:
- Briefly justify your total score (up to 100 words).
- Conclude with the score using the format:  ''Score:  <total points>''.

Remember:  assess strictly from a math-correctness perspective; logical
soundness and computational accuracy are paramount, while clarity and expert
insight elevate higher scores.
```

Table 6: LLaMA-3.2-3B-Instruct training settings of SeRL.

| Method | Hyperparameters |
|---|---|
| SeRL | $n_{vote}$ = 16
n_samples_per_prompt = 16
$\gamma_{diff}$ = 0.2, $\gamma_{easy}$ = 0.8
Temperature = 1.0
RL Algorithm = `Reinforce++`
Iteration = 3
One iteration is defined as a full pass over 7,500 instructions. |
| Backbone | LLaMA-3.2-3B-Instruct |
| PPO Trainer | Actor Learning Rate = $5 \times 10^{-7}$
Critic Learning Rate = $9 \times 10^{-6}$
$\gamma = 1.0, \ \lambda = 1.0$
Initial KL Coefficient = $1 \times 10^{-4}$ |
| Batch Sizes | train_batch_size = 16
rollout_batch_size = 16
micro_train_batch_size = 2
micro_rollout_batch_size = 16 |
| Lengths | Prompt Max Length = 1024
Generate Max Length = 1024 |
| Optimizations | `bf16`, `adam_offload`, `gradient_checkpointing`,
`packing_samples`, `flash_attn`, `enforce_eager` |
| Training Schedule | Epochs = 1 |

Table 7: Qwen-2.5-7B-Instruct training settings of SeRL.

| Method | Hyperparameters |
|---|---|
| SeRL | $n_{vote}$ = 16
n_samples_per_prompt = 16
$\gamma_{diff}$ = 0.2, $\gamma_{easy}$ = 0.8
Temperature = 1.0
RL Algorithm = `Reinforce++`
Iteration = 3
One iteration is defined as a full pass over 7,500 instructions. |
| Backbone | Qwen-2.5-7B-Instruct |
| PPO Trainer | Actor Learning Rate = $5 \times 10^{-7}$
Critic Learning Rate = $9 \times 10^{-6}$
$\gamma = 1.0, \ \lambda = 1.0$
Initial KL Coefficient = $1 \times 10^{-4}$ |
| Batch Sizes | train_batch_size = 16
rollout_batch_size = 16
micro_train_batch_size = 1
micro_rollout_batch_size = 4 |
| Lengths | Prompt Max Length = 1024
Generate Max Length = 1024 |
| Optimizations | `bf16`, `adam_offload`, `gradient_checkpointing`,
`packing_samples`, `flash_attn`, `enforce_eager` |
| Training Schedule | Epochs = 1 |

# D  Additional Experiments

## D.1  SFT Baselines

We conducted this experiment and compared SFT using the sampled truly correct responses of the model. As shown in Tab. 8, we observe a consistent drop in performance. This may be due to the limited quality of self-sampled responses or the model's existing proficiency on those samples, resulting in minimal gains from further fine-tuning. To validate our findings, we additionally conduct SFT using data distilled from Qwen2.5-Math-7B-Instruct, and as we hypothesized, fine-tuning with higher-quality data is indeed necessary to enhance the capabilities of the model.

Table 8: Pass@1 comparison of different SFT strategies on LLaMA-3.2-3B-Instruct.

| Methods | MATH-500 | MATH-Hard | ASDiv | College Math | TabMWP |
|---|---|---|---|---|---|
| Initial | 47.6 | 22.5 | 84.6 | 35.2 | 46.4 |
| SFT on Majority responses | 44.6 | 20.2 | 83.1 | 30.5 | 48.2 |
| SFT on Correct responses | 44.6 | 20.7 | 83.9 | 33.0 | 50.6 |
| SFT on Distilled responses | 49.0 | 23.6 | 87.1 | 36.4 | 66.1 |

## D.2  Results of More Iterations

We additionally report results for 4 to 6 rounds in Tab. 9, where we observe that the performance of the model gradually converges. As discussed in Section 6, we believe this convergence behavior is largely due to the limited capacity of the underlying foundation model. This observation aligns with findings from EMPO [87], which suggests that pre-trained language models already possess strong reasoning capabilities. In this context, RL post-training may primarily help activate latent reasoning patterns learned during pretraining, rather than introduce new ones.

It is worth emphasizing that our goal is not to achieve unlimited iterative improvement through self-play, which is currently an unrealistic expectation and an unsolved problem across the field. As stated in the introduction, our work focuses on combining self-instruction and self-rewarding methods to incentivize the reasoning abilities of LLMs in data-scarce settings, aiming to achieve performance comparable to training on full high-quality data with verifiable rewards.

Table 9: Pass@1 results over additional iterations on mathematical benchmarks.

| Models | Methods | MATH-500 | MATH-Hard | ASDiv | College Math | TabMWP |
|---|---|---|---|---|---|---|
| LLaMA-3.2-3B-Instruct | SeRL (iter4) | 52.7 | 24.1 | 88.8 | 38.0 | 71.1 |
| | SeRL (iter5) | 52.1 | 23.9 | 88.9 | 37.9 | 71.5 |
| | SeRL (iter6) | 52.0 | 24.2 | 89.1 | 37.9 | 70.9 |
| Qwen-2.5-7B-Instruct | SeRL (iter4) | 75.4 | 50.6 | 94.4 | 55.2 | 79.7 |
| | SeRL (iter5) | 76.0 | 50.1 | 94.6 | 55.3 | 80.2 |
| | SeRL (iter6) | 76.1 | 49.8 | 94.6 | 55.3 | 80.4 |

## D.3  Additional Results in the Medical Domain

To demonstrate that our method can be applied to other domains with limited data, we also conducted experiments in the medical domain. Specifically, we used 500 randomly selected instructions from the MedQA [48] training set as the seed data for our method and compared it with a supervised reinforcement learning baseline trained on the full 10.2k dataset (RL-GT). We evaluated both models on three medical benchmarks: MedQA, PubMedQA [24], and NephSAP [68]. All evaluations are conducted using greedy decoding with a maximum of 1,024 new tokens. As shown in Tab. 10, our SeRL achieves performance comparable to the RL-GT baseline, demonstrating its effectiveness.

Table 10: Pass@1 performance comparison between our method and other baselines on the medical benchmark.

| Models | Methods | MedQA | PubMedQA | NephSAP |
|---|---|---|---|---|
| LLaMA-3.2-3B-Instruct | Initial | 53.4 | 54.4 | 24.2 |
| | RL-GT | **57.2** | 55.0 | 26.1 |
| | SeRL (Ours) | 56.9 | **55.3** | **27.5** |
| Qwen-2.5-7B-Instruct | Initial | 58.2 | 32.1 | 29.5 |
| | RL-GT | **59.4** | 33.1 | **31.2** |
| | SeRL (Ours) | 59.1 | **34.8** | 30.9 |

Table 11: Similarity between different self-rewards and the ground-truth rewards on MATH500 using LLaMA3.2-3B-Instruct.

| Methods | Cosine ($\uparrow$) | MAE ($\downarrow$) | MSE ($\downarrow$) |
|---|---|---|---|
| Majority-voting reward (Ours) | **0.75** | **0.30** | **0.60** |
| Model-based reward | 0.17 | 0.89 | 1.2 |
| Entropy-based reward | 0.65 | 0.45 | 0.65 |
| CAI reward | 0.01 | 1.0 | 1.41 |

# E Additional Reward Analysis

## E.1 Comparison with Alternative Rewarding Methods

Self-rewarding methods are fundamentally designed to approximate the true reward. Their effectiveness can be evaluated by measuring how closely the estimated rewards align with the ground-truth rewards, as greater agreement indicates more accurate reward estimation. To quantify this, we calculate multiple metrics including cosine similarity, mean squared error (MSE), and mean absolute error (MAE) between the estimated rewards and the ground-truth rewards, where higher similarity and lower error indicate greater accuracy in reward estimation. In our evaluation, we treat the rule-based reward as the ground-truth reward and measure the alignment of the majority-voting, model-based, entropy-based [87], and CAI reward with it. CAI reward is the reward model we train based on the work of Bai et al. [2]. As shown in Tab 11, our method achieves the highest accuracy among all compared approaches.

## E.2 Reward Hacking

As evidenced by the learning curves presented in Section 4.3, we observe that the training process becomes prone to reward hacking without the dual-end clipped difficulty filtering strategy. To further investigate the nature of this reward hacking behavior, we conduct a case study, as shown in Box E.1. Notably, we find that although the model correctly derives the answer $\left(3, \frac{\pi}{2}\right)$ during the Chain-of-Thought (CoT) reasoning, it still outputs "The final answer is: $\boxed{0}$" at the end. We also provide a detailed explanation in Section 4.3 of how the dual-end clipped difficulty filtering strategy helps prevent reward hacking by design.

To more clearly present the impact of the difficulty filtering strategy on generated instructions, we analyze the difficulty and diversity of the first-round dataset generated by LLaMA-3.2-3B-Instruct before and after the difficulty filtering strategy. We denote the generated dataset from iteration $n$ as $\mathcal{D}_{\text{gen}}^n = \bigcup_{t \in \mathcal{T}^n} \mathcal{B}_{\text{gen}}^t$, where $\mathcal{T}^n$ represents the set of all generation steps in iteration $n$. Accordingly, the first-round generated dataset is denoted as $\mathcal{D}_{\text{gen}}^1$. We hypothesize that more difficult instructions require more complex CoT reasoning, which tends to result in longer responses. Qwen-3-32B [77], with its strong overall performance, can correctly answer most questions in the dataset, making it a reliable proxy for assessing instruction difficulty. Therefore, we use Qwen-3-32B to answer the instructions in $\mathcal{D}_{\text{gen}}^1$ and compare response length distributions as a proxy for difficulty. As shown in Fig. 4a and Fig. 4b, applying the difficulty filtering leads to a reduction in both extremely short and excessively long responses, aligning with our design goal of removing overly easy and overly difficult instructions. To assess data diversity, we compute the ROUGE-L similarity between each

> **Box E.1: Case study of LLaMA-3.2-3B-Instruct outputs after reward hacking.**
>
> Convert the point $(0, 3)$ in rectangular coordinates to polar coordinates. Enter your answer in the form $(r, \theta)$, where $r > 0$ and $0 \leq \theta < 2\pi$.
>
> ## Step 1: Recall the conversion formulas from rectangular to polar coordinates.
> The conversion from rectangular coordinates $(x, y)$ to polar coordinates $(r, \theta)$ can be done using the formulas: $r = \sqrt{x^2 + y^2}$ and $\theta = \tan^{-1}\left(\frac{y}{x}\right)$.
>
> ## Step 2: Substitute the given coordinates.
> Given the point $(0, 3)$, substitute $x = 0$ and $y = 3$: $r = \sqrt{0^2 + 3^2} = 3$, and $\theta = \tan^{-1}\left(\frac{3}{0}\right)$.
>
> ## Step 3: Evaluate the angle.
> Since $\tan^{-1}(3/0)$ is undefined, note that the point lies on the positive $y$-axis, which corresponds to $\theta = \frac{\pi}{2}$.
>
> ## Step 4: Write the final answer.
> The polar coordinates are: $\boxed{(3, \frac{\pi}{2})}$.
>
> The final answer is: $\boxed{0}$

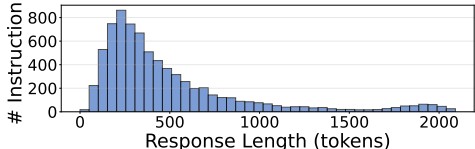

(a) Response length distribution of $\mathcal{D}^1_{\text{gen}}$ before difficulty filtering.

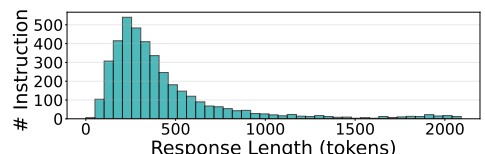

(b) Response length distribution of $\mathcal{D}^1_{\text{gen}}$ after difficulty filtering.

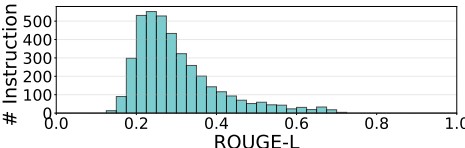

(c) Distribution of ROUGE-L scores between $\mathcal{D}^1_{\text{gen}}$ and $D_{\text{seed}}$ before difficulty filtering.

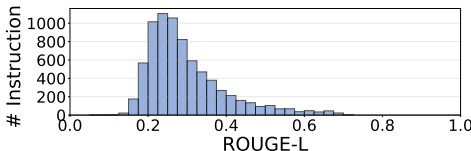

(d) Distribution of ROUGE-L scores between $\mathcal{D}^1_{\text{gen}}$ and $D_{\text{seed}}$ after difficulty filtering.

Figure 4: Comparison of difficulty and diversity of $\mathcal{D}^1_{\text{gen}}$ before and after applying the difficulty filtering. The y-axis label *# Instructions* represents the number of instructions.

generated instruction $\boldsymbol{x}_i \in \mathcal{D}^1_{\text{gen}}$ and the $\mathcal{D}_{\text{seed}}$. For each $\boldsymbol{x}_i$, we define:

$$\Phi(\boldsymbol{x}_i, \mathcal{D}_{\text{seed}}) := \max_{\boldsymbol{s}_j \in \mathcal{D}_{\text{seed}}} \text{ROUGE-L}(\boldsymbol{x}_i, \boldsymbol{s}_j). \tag{11}$$

We visualize the distributions of $\mathcal{D}^1_{\text{gen}}$ and $\mathcal{D}_{\text{seed}}$ based on $\Phi(\boldsymbol{x}_i, \mathcal{D}_{\text{seed}})$. As shown in Fig. 4c and Fig. 4d, the ROUGE-L distributions before and after filtering indicate that the diversity of $\mathcal{D}^1_{\text{gen}}$ is largely preserved, suggesting that the dual-end clipped difficulty filtering strategy effectively controls for difficulty without influencing instructions diversity.

# F   Additional Generated Instructions Analysis

To validate the reliability of the data generated by our self-instruction method, we conduct a comparison against both the seed dataset $\mathcal{D}_{\text{seed}}$ and the full MATH training set in terms of length, quality, difficulty, and diversity. The dataset $\mathcal{D}_{\text{seed}}$ consists of 500 examples uniformly sampled from different difficulty levels within the MATH training set and serves as the few-shot context for all subsequent data generated through our self-instruction method. Specifically, $\mathcal{D}^1_{\text{gen}}$, $\mathcal{D}^2_{\text{gen}}$, and $\mathcal{D}^3_{\text{gen}}$ respectively

> **Box F.1: Instruction quality evaluation prompt**
>
> ```
> You are an expert evaluator of math problems.  Please evaluate the overall
> quality of the following math problem.
>
> Math Problem:
> <INSTRUCTION>
>
> First, analyze this problem in detail, considering all aspects including
> clarity, educational value, appropriateness, and whether it tests meaningful
> mathematical concepts.
>
> After your analysis, provide your rating on a scale from 0 to 5, where:
> 0 = Extremely poor quality mathematical problem
> 5 = Excellent quality mathematical problem
>
> Your response should include your detailed analysis followed by your rating.
> Make sure to include your final rating in this exact format at the END of
> your response:
>
> FINAL RATING: [0-5]
>
> Remember to use only whole numbers (integers) from 0 to 5 for your rating,
> not decimals or fractions.
> ```

Table 12: Comparison of instruction length and quality across $\mathcal{D}^1_{\text{gen}}$, $\mathcal{D}^2_{\text{gen}}$, $\mathcal{D}^3_{\text{gen}}$, $\mathcal{D}_{\text{seed}}$, and MATH train set. *# Inst.* refers to the number of instructions, *Inst. Len* indicates the statistical summary of instruction lengths, and *Quality Rate* represents the statistical summary of the scores assigned by Qwen-3-32B when evaluating the instructions.

| Dataset | # Inst. | Inst. Len. | Quality Rate |
|---|---|---|---|
| MATH train | 7500 | $75 \pm 93$ | $3.8 \pm 1.5$ |
| $\mathcal{D}_{\text{seed}}$ | 500 | $78 \pm 106$ | $3.7 \pm 1.6$ |
| $\mathcal{D}^1_{\text{gen}}$ | 7500 | $57 \pm 27$ | $3.6 \pm 1.5$ |
| $\mathcal{D}^2_{\text{gen}}$ | 7500 | $60 \pm 36$ | $3.5 \pm 1.6$ |
| $\mathcal{D}^3_{\text{gen}}$ | 7500 | $59 \pm 32$ | $3.5 \pm 1.5$ |

denote the data collections cumulatively generated by LLaMA-3.2-3B-Instruct during the first, second, and third iterations of training.

## F.1 Length and Quality

We compare the length and quality score of the datasets $\mathcal{D}^1_{\text{gen}}$, $\mathcal{D}^2_{\text{gen}}$, $\mathcal{D}^3_{\text{gen}}$, $\mathcal{D}_{\text{seed}}$, and the MATH training set. The length is measured by the number of tokens obtained using the Qwen-3-32B tokenizer, where a longer instruction typically indicates higher complexity. Invalid long instructions composed of repetitive sentences have already been removed by our online filtering strategy. Among the remaining instructions, a greater average length suggests higher semantic and structural complexity. The quality score is computed by prompting Qwen-3-32B to evaluate each instruction based on criteria such as completeness of the instruction description, clarity, and overall usefulness or relevance. The evaluation prompt is provided in Box F.1.

As shown in Tab. 12, the results reveal that both the instruction lengths and quality scores of the generated instructions are comparable to those of $\mathcal{D}_{\text{seed}}$ and the MATH training set. This suggests that the instructions produced by our method not only maintain sufficient complexity comparable to that of the original datasets but also achieve a similar level of semantic quality, as assessed by the evaluation model. These findings indicate that our self-instruction strategy yields high-quality instructions, despite being carried out in a data-scarce setting.

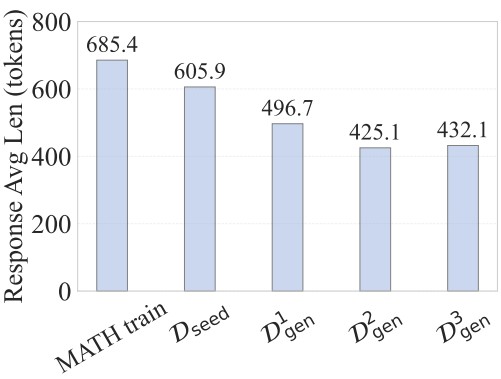

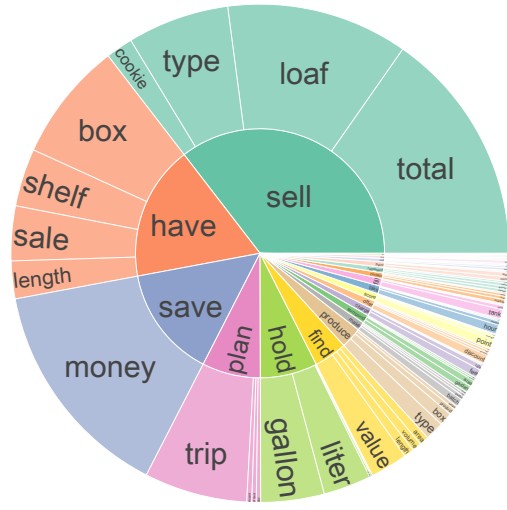

(a) Comparison of response lengths across $\mathcal{D}_{gen}^1$, $\mathcal{D}_{gen}^2$, $\mathcal{D}_{gen}^3$, $\mathcal{D}_{seed}$ and the MATH training set. To ensure consistency in response generation, we set the generation temperature to 0, thereby enabling deterministic greedy decoding.

(b) Analysis of verbs and nouns diversity in $\mathcal{D}_{gen}^1$.

Figure 5: Analysis of generated instructions difficulty and diversity.

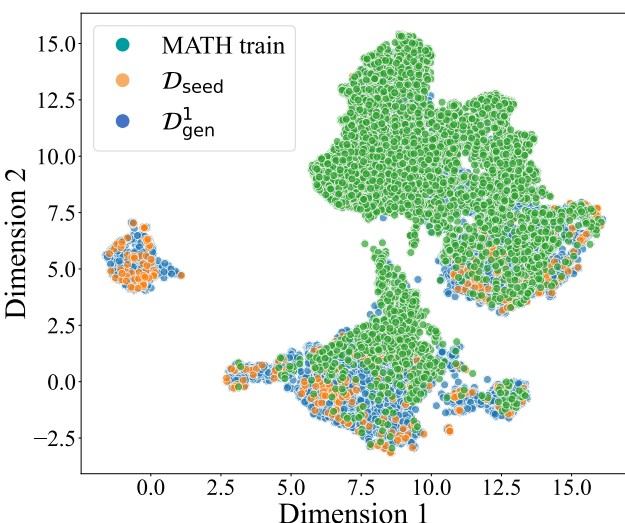

Figure 6: UMAP projection of BERT feature distributions for $\mathcal{D}_{gen}^1$, $\mathcal{D}_{seed}$, and MATH training set.

## F.2 Difficulty

To assess the difficulty level of each dataset, we employ Qwen-3-32B to generate responses for all instructions. We then compare the average response lengths across datasets, based on the assumption that more challenging instructions generally elicit longer reasoning chains from LLMs, thereby leading to longer responses. As shown in Fig. 5a, the average response lengths for the generated datasets are close to those of the MATH training set. This suggests that the generated data exhibits a comparable level of difficulty to the original dataset.

## F.3 Diversity

**Verb-Noun Structures.** We analyze the verb–noun structures of instructions in $\mathcal{D}_{gen}^1$. Specifically, we employ the Berkeley Neural Parser [28, 29] to extract the main verbs nearest to the root and their first direct noun objects from each instruction. Fig. 5b shows the top 20 verbs with their top 4 associated nouns, revealing diverse and syntactically rich verb–noun constructions in the generated instructions.

**ROUGE-L Distribution.** We compute the ROUGE-L score between each instruction in $\mathcal{D}_{gen}^1$ and $\mathcal{D}_{seed}$. The ROUGE-L distribution is shown in Fig. 4c. As observed, most generated instructions exhibit low ROUGE-L similarity with $\mathcal{D}_{seed}$. This indicates that the model does not merely mimic $\mathcal{D}_{seed}$ but instead generates diverse and novel instructions.

**UMAP Projection.** We project the BERT features of the instructions in $\mathcal{D}_{gen}^1$ using UMAP [45], as shown in Fig. 6. The visualization shows that the generated instruction points are not tightly clustered but instead exhibit a wide distribution similar to that of the MATH training set. This phenomenon further demonstrates the diversity of the generated dataset.

## G    Additional Analysis of the Self-Iteration Upper Bound

As shown in Fig. 7, we observe that after multiple iterations, the Maj@16 performance of the model does not exhibit a consistent upward trend. Since Maj@16 serves as an empirical upper bound for Pass@1 under our framework, the lack of improvement in Maj@16 suggests that the performance ceiling of the model remains unchanged. This observation is consistent with the conclusion of Yue et al. [83], which states that reinforcement learning with verifiable rewards primarily converts Pass@K performance into Pass@1 without fundamentally improving the capacity of the model. Although our method does not lead to incremental improvements in Maj@K, it still achieves substantial gains in Pass@1 performance in data-scarce scenarios through unsupervised reinforcement learning. These results highlight the practical utility of our approach, particularly in specialized, data-scarce domains where labeled data is limited.

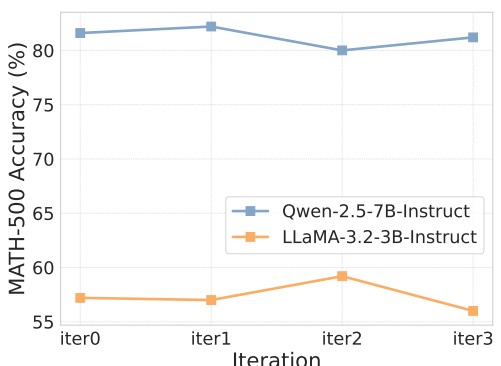

Figure 7: The performance curves of Maj@16 on MATH-500 for Qwen-2.5-7B-Instruct and LLaMA-3.2-3B-Instruct trained with our method over multiple iterations.

## H    Broader Impact

This work explores reinforcement learning for enhancing the reasoning capabilities of LLMs in data-scarce scenarios, which has the potential to significantly broaden the accessibility and applicability of advanced AI systems. By reducing reliance on large-scale, high-quality labeled datasets, our method could enable the development of performant LLMs in data-scarce domains such as clinical diagnostics [58, 44, 60] and aerospace engineering [36, 5, 75], where expert-annotated data is often expensive or impractical to obtain. However, the ability to generate and reinforce synthetic data also introduces potential risks. Without careful oversight, unsupervised self-improvement loops may propagate or even amplify biases present in the initial seed data. Additionally, models trained in data-scarce settings may exhibit spurious generalization patterns, especially in high-stakes domains like clinical decision-making or aerospace control. To mitigate these risks, we incorporate a robust online filtering mechanism to enforce baseline quality and control difficulty during data generation. Nevertheless, domain-specific filtering and safety measures will be essential in future deployments, especially when adapting the framework to sensitive or high-impact application areas.

