# OpenReview forum: "SeRL: Self-play Reinforcement Learning for Large Language Models with Limited Data"
_NeurIPS.cc/2025/Conference — NeurIPS 2025 poster_

### Official Review · Reviewer_MF7H · 2025-06-30

**Clarity:** 2
**Significance:** 3
**Originality:** 3
**Rating:** 4
**Confidence:** 3

**Summary:**

This paper proposes Self-Play Reinforcement Learning (SeRL) which is a novel approach to bootstrap the LLM training with limited data. It contains two modules called self-instruction and self-rewarding which are used to generate the instructions and assign the response rewards for this additional data. SeRL leverages self-instruction and self-rewarding to increase the limited data during the post-training of the LLMs.
The empirical results show that SeRL outperforms the advanced self-play methods.

**Questions:**

- Can you elaborate more on the Online Instruction Filter? Can you provide more results on the various criteria used in the filtering?
- Why use the cosine similarity for evaluating the self-rewarding?
- I am interested in Table 4. Why does the best performance appear in iterations 1-3 for the three RL methods? And why the Reinforce++ show the most stable and robust performance?
- I am curious how the difficulty filtering is adopted. The reward is assigned with 0 or 1 for the response. How do you determine the ratio of the difficulty filtering from 0.2 to 0.8?

**Ethical Concerns:**

["NO or VERY MINOR ethics concerns only"]

**Final Justification:**

The authors' rebuttal solves my questions. Regarding the novelty of the idea and the current score, I prefer to maintain the score.

**Limitations:**

Yes.

**Paper Formatting Concerns:**

No formatting concerns.

**Quality:**

3

**Strengths And Weaknesses:**

**Strengths:**

**Quality.** The paper provides a self-play method to enlarge the dataset for the LLMs post-training. The self-instruction is filtered by several criteria to ensure the quality of the generated instructions. In the self-rewarding module, a vote-based reward is used. The ablation study supports the claim of the effectiveness of the self-instruction and self-rewarding modules.

**Clarity.** The paper is well-structured in the presentation of the methodology and experiments.

**Significance.** In practice, it is hard to collect a large amount of data for the LLMs post-training in reinforcement learning. The proposed method provides a way to enlarge the dataset for the LLMs post-training.

**Originality.** The self-rewarding with the vote-based mechanism is novel and derived from the recent proposed work with the 'RL with verifiable labels can be viewed as converting a model’s Pass@K performance into Pass@1'.


**Weaknesses:**

**Quality.** The paper provides the empirical results showing that SeRL outperforms the existing methods and two ablation studies to demonstrate the effectiveness of the self-instruction and self-rewarding modules. However, the first ablation study only demonstrates the difficulty filter on LLaMA. It should contain a comprehensive analysis of all filtering criteria.

**Clarity.** The Online Instruction Filter is not very clear and should be elaborated with details.

**Significance.** The paper provides a novel method to enlarge the dataset for the LLMs post-training but lacks methods to mitigate the potential bias introduced by the self-instruction and self-rewarding modules, which limits the practical deployment of the method.

---

> ### Author Rebuttal · Authors · 2025-07-31
>
> Dear Reviewer MF7H,
>
> Thanks for your professional review! We sincerely appreciate your recognition of the strengths of our work: (1) the effectiveness of our self-instruction and self-rewarding modules, (2) the practical value in enlarging the dataset for RL training, (3) the novelty of our majority-vote-based self-rewarding method, and (4) the clear and well-structured presentation of our methodology and experiments.
>
> **[W1] Demonstrate a comprehensive analysis of all filtering criteria.**
>
> Thanks for the constructive suggestions. As suggested, we have additionally conducted ablation studies to demonstrate the effectiveness of each filtering strategy. The results in Table R1 show that the effectiveness of each filtering strategy, as removing any of them leads to a decline in the overall performance of the model. In particular, omitting the difficulty filter may result in reward hacking, as illustrated in Figure 2 of the original manuscript.
>
> Table R1: Pass@1 comparison under the same number of training steps with different filtering strategies ablated, using LLaMA-3.2-3B-Instruct.
>
> | Methods | MATH-500 | MATH-Hard | ASDiv | College Math | TabMWP |
> | - | - | - | - | - | - |
> | SeRL | 52.6 | 23.7 | 89.0 | 37.7 | 70.6 |
> | SeRL w/o Length Filter | 47.6 | 23.2 | 87.4 | 36.2 | 60.1 |
> | SeRL w/o Keywords Filter | 48.0 | 22.1 | 87.4 | 36.3 | 62.5 |
> | SeRL w/o Similarity Filter | 48.8 | 23.3 | 87.3 | 35.6 | 64.6 |
> | SeRL w/o Difficulty Filter | 11.6 | 5.1 | 1.5 | 14.0 | 10.2 |
>
> ---
>
> **[W2\&Q1] More details on the online instruction filter, and additional results for individual filtering criteria.**
>
> Sorry for the confusion. The proposed online instruction filter consists of four components:
>
> - (1) The similarity filter computes the ROUGE-L score between a newly generated instruction and existing instructions. If the score exceeds a predefined threshold (set to 0.7 following the prior work [1]), indicating high similarity to existing instructions, the new instruction is filtered out to encourage diversity.
> - (2) The keywords filter removes instructions containing specific keywords such as "image", "graph", "picture", "file", "map", "draw", "plot", or "write a program", as they refer to visual content or capabilities beyond the model's scope. In addition, instructions starting with punctuation or non-English characters are also excluded.
> - (3) The length filter removes instructions that are either excessively long or short. Instructions exceeding 150 words often contain redundant content or even include solutions, while those with fewer than 3 words typically lack sufficient context for problem solving. This filtering step helps maintain the overall quality and clarity of the generated instructions.
> - (4) The difficulty filter evaluates the proportion of majority answers among the generated responses for a given instruction. Instructions with excessively high or low majority proportions are filtered out to ensure that the retained instructions maintains suitable difficulty for model learning. We set the thresholds as $\gamma_{\text{difficult bound}} = 0.2$ and $\gamma_{\text{easy bound}} = 0.8$, as shown in Table 1 of the original supplementary material.
>
> ---
>
> **[W3] Methods to mitigate bias from the self-instruction and self-rewarding modules.**
>
> Thanks for the insightful comments. Our proposed method is specifically designed to minimize this bias as much as possible, and matches the performance of methods using extensive data with verifiable rewards, dispite the limited amount of seed data used in our method.
>
> (1) For the self-instruction module, we introduce an online filtering strategy that maintains data quality while promoting diversity in the generated samples. We have provided a comprehensive analysis of the generated data in terms of quality, difficulty, and diversity, as detailed in Appendix E of the original supplementary material. These results demonstrate the reliability of our self-instruction method for data generation.
>
> (2) For the self-rewarding module, as stated in the related works [2-3, 7], estimation bias is inevitable in self-rewarding settings. Our majority-voting method assigns rewards based on the consistency among multiple sampled responses, offering greater stability than scoring individual responses. To demonstrate this, we have additionally conducted experiments with more self-rewarding baselines. The results in Table R1 show that our majority-vote-based reward method achieves higher accuracy, indicating lower bias compared to other self-rewarding methods.
>
> Finally, our proposed SeRL can achieve performance on par with those obtained by verifiable ground-truth rewards (RL-GT baseline), as shown in Table 1 of the original manuscript.
>
> Table R2: Similarity between different self-rewards and the ground-truth rewards on MATH500 using LLaMA3.2-3B-Instruct. We assess the alignment between self-rewards and ground-truth rewards of all responses to MATH500 by treating them as vectors and computing similarity metrics; see Figure 3 of the original manuscript for details.
>
> | Methods | Cosine (↑) | MAE (↓) | MSE (↓) |
> | - | - | - | - |
> | Majority-vote-based reward (Ours) | **0.75** | **0.30** | **0.60** |
> | Model-based reward [1] | 0.17 | 0.89 | 1.2 |
> | Entropy-based reward [3] | 0.65 | 0.45 | 0.65 |
> | CAI reward [4] | 0.01 | 1.0 | 1.41 |
>
> ---
>
> **[Q2] Why use the cosine similarity for evaluating the self-rewarding?**
>
> Thanks for your comment. The relative accuracy rankings of different self-rewarding methods remain consistent across various similarity metrics, so we chose the simple and intuitive cosine similarity. We additionally report results using MAE and MSE as evaluation metrics, as shown in Table R2.
>
> ---
>
> **[Q3] I am interested in Table 4. Why does the best performance appear in iterations 1-3 for the three RL methods? And why the Reinforce++ show the most stable and robust performance?**
>
> Thanks for the valuable feedback.
>
> (1) To ensure a fair comparison, we follow the same three-round training setup as self-rewarding baseline [2]. We have additionally reported results for 4 to 6 rounds in Table R1, where we observe that the performance of the model gradually converges. As discussed in Section 6 of the original manuscript, we believe this convergence behavior is largely due to the limited capacity of the underlying foundation model. This observation aligns with findings from EMPO [3], which suggests that pre-trained language models already possess strong reasoning capabilities. In this context, RL post-training may primarily help activate latent reasoning patterns learned during pretraining, rather than introduce new ones.
>
> (2) Reinforce++ demonstrates greater robustness compared to GRPO and RLOO due to the following reasons:
>
> - GRPO normalizes the estimated advantage over the $n_{\text{vote}}$ responses for each question, whereas Reinforce++ estimates the advantage over all responses across all questions, resulting in a larger group and more stable estimation. In addition, GRPO uses an external $k_3$ KL estimation, which involves an exponential term. This may cause large spikes in the gradient, leading to instability during training.
> - RLOO applies a preprocessing step that subtracts the average reward of other responses from the reward of each individual response, thereby increasing the reward gap between different responses. However, since self-rewarding may introduce inaccurate reward estimations, the amplified reward differences in RLOO could lead to greater bias when the estimation is incorrect. As a result, its performance tends to be worse than that of Reinforce++.
>
> Table R4: Pass@1 results over multiple iterations on mathematical benchmarks.
>
> | Models | Methods | MATH-500 | MATH-Hard | ASDiv | College Math | TabMWP |
> | - | - | - | - | - | - | - |
> | LLaMA3.2-3B-Instruct | SeRL (iter1) | 48.6 | 23.0 | 87.5 | 36.7 | 68.4 |
> | | SeRL (iter2) | 50.4 | 23.6 | 88.9 | **38.2** | **72.3** |
> | | SeRL (iter3) | 52.6 | 23.7 | 89.0 | 37.7 | 70.6 |
> | | SeRL (iter4) | **52.7** | 24.1 | 88.8 | 38.0 | 71.1 |
> | | SeRL (iter5) | 52.1 | 23.9 | 88.9 | 37.9 | 71.5 |
> | | SeRL (iter6) | 52.0 | **24.2** | **89.1** | 37.9 | 70.9 |
> | Qwen2.5-7B-Instruct | SeRL (iter1) | 74.2 | 50.0 | 94.2 | 54.3 | 77.0 |
> | | SeRL (iter2) | 74.8 | 49.7 | **94.7** | 55.1 | 80.1 |
> | | SeRL (iter3) | 75.8 | 50.4 | 94.4 | 55.1 | 79.4 |
> | | SeRL (iter4) | 75.4 | **50.6**  | 94.4 | 55.2 | 79.7 |
> | | SeRL (iter5) | 76.0 | 50.1 | 94.6 | **55.3** | 80.2 |
> | | SeRL (iter6) | **76.1** | 49.8 | 94.6 | **55.3** | **80.4** |
>
> ---
>
> **[Q4] I am curious how the difficulty filtering is adopted. The reward is assigned with 0 or 1 for the response. How do you determine the ratio of the difficulty filtering from 0.2 to 0.8?**
>
> Thanks for your insightful questions. As described in the original manuscript (Page 4, Lines 130-135), the majority answer among different sampled responses is treated as the ground truth. For a given question, let $n_{\text{vote}}$ be the total number of responses and $n_{\text{maj}}$ the number of responses agreeing with the majority answer. We define the ratio (average accuracy) as  $n_{\text{maj}} / n_{\text{vote}}$. Finally, we filter the instructions whose ratios fall outside the difficulty range [0.2, 0.8].
>
> ---
>
> Lastly, thanks to your constructive feedback. If you have further questions, please feel free to let us know.
>
> **References**
>
> [1] Self-Instruct: Aligning Language Models with Self-Generated Instructions. arXiv 2022.
>
> [2] Self-Rewarding Language Models. ICML 2024.
>
> [3] Right Question is Already Half the Answer: Fully Unsupervised LLM Reasoning Incentivization. arXiv 2025.
>
> [4] Constitutional AI: Harmlessness from AI Feedback. arXiv 2022.
>
> [5] WizardLM: Empowering large pre-trained language models to follow complex instructions. arXiv 2023.
>
> [6] TeaMs-RL: Teaching LLMs to Generate Better Instruction Datasets via Reinforcement Learning. arXiv 2024.
>
> [7] TTRL: Test-Time Reinforcement Learning. arXiv 2025.

---

> > ### Comment · Reviewer_MF7H · 2025-08-06
> >
> > Thank you for your responses. And it solves my most concerns and I have one more question on **W1**.
> >
> > ---
> > **W1**: The results show that filter criteria, except for the difficulty criterion, have a limited impact on model performance. Removing these criteria does not lead to any significant performance degradation. Can you summarize the results and make an explanation of Table R1?

---

> ### Comment · Area_Chair_d8Lz · 2025-08-05
> **Author-Reviewer Discussions**
>
> Dear Reviewer MF7H,
>
> Thanks for your review! The authors have provided their rebuttal. Please respond to the authors and update your review as appropriate.
>
> Thank you!
> AC

---

> ### Author Response · Authors · 2025-08-07
> **Thanks for the feedback!**
>
> Thank you once again for your constructive review and positive support, which have greatly contributed to improving the clarity and quality of our manuscript. Below, we sincerely address the new points you have raised.
>
> (1) As observed by the reviewer, Table R1 shows that only removing the difficulty filter leads to a significant performance degradation. However, we believe that the removal of other filtering strategies also results in notable accuracy drops, as reflected in the accuracy changes on benchmarks such as MATH-500 and TabMWP.
>
> (2) The difficulty filter is specifically introduced under our self-rewarding setup. It filters out instructions for which the model's responses show low confidence, defined as $n\_{\text{maj}} / n\_{\text{vote}} < \gamma\_{\text{difficulty bound}}$. This directly reduces the bias introduced by the majority-vote-based self-rewarding method and prevents reward hacking, as illustrated in Section 4.3 of the original manuscript, thereby avoiding a significant degradation in model performance.
>
> (3) The other filters are the basic filtering strategies we adopt following Wang et al. [1]. As described in Appendix E of the original supplementary material, these filters help ensure a certain level of instruction diversity and quality, contribute to performance improvements to some extent, and incur low computational cost. Therefore, we choose to retain them.
>
> (4) We further conduct additional experiments to analyze the filtering strength and relationships among different strategies. Specifically, we examine the number of instructions filtered by each strategy to assess their respective filtering strength, and we analyze the overlap among the filtered instructions to understand their interrelations. As shown in Table R5, we find that the difficulty filter has the strongest filtering strength, followed by the similarity filter, while the length and keywords filters have relatively weaker strength. As shown in Table R6, we observe that the difficulty filter also filters the majority of instructions filtered by the other strategies. This can be attributed to the following reasons:
>
> - The length filter and keywords filter are used to enhance the quality of instructions, as low-quality instructions tend to increase the uncertainty in the responses of the model. The difficulty filter is designed to remove instructions that elicit high response uncertainty from the model, thereby indirectly filtering out a portion of low-quality instructions as well.
> - Meanwhile, the similarity filter aims to maintain instruction diversity by filtering out similar instructions. Repeated training on highly similar instructions may lead to overly deterministic responses from the model on such instructions. The difficulty filter also helps exclude instructions that already yield highly deterministic responses, characterized by $n\_{\text{maj}} / n\_{\text{vote}} > \gamma\_{\text{easy bound}}$, thus indirectly contributing to filtering out similar instructions as well.
>
> In summary, the difficulty filter plays a critical role in preventing model collapse due to reward hacking. At the same time, the other strategies complement it by maintaining instruction quality and diversity, collectively contributing to improved model performance.
>
> Table R5: The number of instructions filtered by applying each filtering strategy individually out of 10,000 instructions generated by LLaMA3.2-3B-Instruct.
>
> |                         | Length Filter | Keywords Filter | Similarity Filter | Difficulty Filter |
> | ----------------------- | ------------- | --------------- | ----------------- | ----------------- |
> | # Filtered Instructions | 1212          | 1007            | 3461              | 4578              |
>
> Table R6: The number of overlapping instructions filtered by different strategies out of the 10,000 instructions generated by LLaMA3.2-3B-Instruct. The numbers in the table represent the count of overlapping instructions filtered by the two corresponding strategies. The percentages in parentheses indicate the proportion of these overlapping instructions relative to each individual strategy: the first percentage corresponds to the row strategy, and the second to the column strategy.
>
> |                   | Length Filter | Keywords Filter    | Similarity Filter  | Difficulty Filter   |
> | ----------------- | ------------- | ------------------ | ------------------ | ------------------- |
> | Length Filter     | -             | 404 (33.3%, 40.1%) | 506 (41.7%, 14.6%) | 673 (55.5%, 14.7%)  |
> | Keywords Filter   | -             | -                  | 201 (20.0%, 5.8%)  | 877 (87.1%, 19.2%)  |
> | Similarity Filter | -             | -                  | -                  | 2467 (71.3%, 53.9%) |
> | Difficulty Filter | -             | -                  | -                  | -                   |
>
> **Reference**
>
> [1] Self-Instruct: Aligning Language Models with Self-Generated Instructions. arXiv 2022.

---

### Official Review · Reviewer_dA2g · 2025-07-02

**Clarity:** 3
**Significance:** 3
**Originality:** 4
**Rating:** 5
**Confidence:** 3

**Summary:**

This paper introduces Self-Play Reinforcement Learning (SeRL), a method for training language models with reinforcement learning without any reward labels and with only a very small initial prompt dataset. The authors position this approach as an alternative to reinforcement learning with verifiable rewards, i.e., they aim to improve language model capabilities and reasoning (as opposed to preference-based RL fine-tuning).

The method is based on synthetic data generation on two fronts: (1) The initial set of prompts is successively expanded by asking the current iteration of the LLM in-training to propose new prompts in a few-shot manner (self-instruct). Prompts are generated and then filtered with the aim to generate solvable tasks (e.g., no tasks involving visuals) that are neither too difficult or too easy, as measured by the agreement of language model responses when sampling repeatedly. (2) Responses for all prompts (both the initially provided ones and the self-generated ones) are rewarded based on majority voting, i.e., by sampling multiple responses, assuming the majority response is the ground-truth, and then applying a 0/1 reward.

In their experimental evaluation the authors show that this pipeline can improve language model performance (particularly that of LLaMA-3.2-32B-Instruct and Qwen-2.5-7B-Instruct), reaching and sometimes even exceeding performance of a language model trained on a larger superset of the data with (rule-based) ground-truth rewards. They demonstrate this with a dataset focused on math. Their evaluation also largely focuses on math, but extends to general reasoning. Their method reaches performance of the GT-reward baseline and outperforms baselines with similar iterative self-labeling schemes, but based on DPO instead of scalar-reward RL (I-RPO, SR-DPO).

In their ablation studies the authors further demonstrate the importance of the components of the proposed framework: The difficulty filtering is necessary to avoid reward hacking, the majority-based rewarding is competitive with ground-truth rewards even when tested in isolation without self-instruct, and the framework performance varies with but is reasonably robust to changes in the underlying reinforcement learning algorithm.

The main novelties seem to be the difficulty-based filtering in the self-instruct phase and the majority-vote based self-reward, as well as the iterated online setting.

**Questions:**

Please see the numbered weaknesses, focusing on the major ones (**WX**). Most of them are actionable, i.e., can be considered questions, although many of the minor ones (**MX**) do not require an individual response. Actual questions regarding the understanding of the paper are listed as weaknesses in the clarity section. The minor points have little impact on my score.

I consider W2, W3, and W5 to be of particular importance. Note that W1/W5 can be addressed by clearly acknowledging and discussing the limitation, and do not necessarily require new experiments.

**Ethical Concerns:**

["NO or VERY MINOR ethics concerns only"]

**Final Justification:**

In addition to the strengths raised in my initial review, the experiments in another domain further strengthen the paper. Many of my clarity concerns were resolved, and I believe the remaining ones are minor enough not to stand in the way of acceptance.

**Limitations:**

- **W6** The limitations should be discussed in more depth. Of particular relevance are:

    - The limitation to domains where majority voting is applicable.

    - The evaluation on only a single domain.

    - The risk of collapsing to a single response, as both policy and rewards are iteratively updated.

    - Evaluation with only one seed.

    - The risk to reinforce existing biases, particularly in domains that are not as clear-cut as math (briefly discussed in appendix G).

**Paper Formatting Concerns:**

The concerns below are listed as a service to you; they do not impact my score.

- I cannot discern any ordering in the bibliography. That makes it hard to find an entry when working with a printed copy.
- 42, 291: Author name is in all caps.
- 59: for [the] LLM
- 67: LLMs
- 69: The "reasoning abilities" sentence seems to be misplaced here, it does not fit into the flow of text.
- 75, 186: Should be a parenthetical citation
- 218: inconsistent title case
- Figure 3: Reward should be lower case
- 373: Superfluous comma in bibliography
- Appendix: Prompt listings are labeled as tables.

**Quality:**

3

**Strengths And Weaknesses:**

In the following, I discuss strength and weaknesses on the dimensions of quality, clarity, significance, and originality.
Weaknesses are marked with a **W**, minor weaknesses with an **M**, notes that do not impact my score with **N**, and strengths with an **S**.
Weaknesses are numbered for reference purposes.

## Quality

- **S**: The submission is technically sound, all claims are supported by experimental evidence. The methods, metrics, and benchmarks used for evaluation are appropriate. The paper is complete without obvious missing parts.
- **W1** The empirical evaluation could be more extensive, particularly covering domains beyond math. The authors evaluate on one general reasoning task, but train exclusively on math. This is in slight conflict with the motivation, which mainly discusses hard-to-verify domains beyond math.
- **W2** Is there any reason you could not do supervised fine-tuning on the majority response instead? If not, this would be a natural baseline to compare against.

## Clarity

- **S** The paper is generally well-written and well-organized.
- **W3** How exactly does your chain-of-thought approach work? As far as I can tell that is not described in the paper.
- **W4** It is not entirely clear what exactly is meant by rule-based and model-based rewards.
- **M1** The explanation of REINFORCE++ is not entirely clear, would benefit from an appendix (e.g., how exactly is the advantage computed?)
- **M2** The background on self-instruction (line 81) reads like method, not background. It would be valuable to discuss what the actual background is.
- **M3** 104: Details on how few-shot is selected are missing.
- **M4** 109: Details on filtering are missing. For example, which words are filtered?
- **M5** 117 Can you elaborate on why this would result in zero advantage?
- **M6** 119 Can you give a reference on the vanishing gradients?
- **M7** 171 How does the number of gradient steps in your approach compare to the baselines?
- **M8** 213 Why does RL-GT improve other but SeRL does not?
- **M9** It would be good to have the "with clipped" variant in Figure 2 for comparison.
- **M10** It would be good to see more training plots like Figure 2 in addition to the final results, to see if performance saturates.
- **M11** 291 Details on self instruct are lacking. This paragraph should "compare and contrast" better.
- **M12** The self-rewarding methods paragraph does not discuss how all the listed approaches compare to the proposed approach.
- **N** It is a bit hard to understand what underline in the tables mean, especially as the meaning differs between the two tables.
- **N** I think it is not entirely correct to call the observed behavior reward hacking, since the reward itself is modified. It seems like a sort of collapse.
- **N** The explanation of the return on line 78 is confusing; similar for advantage on line 80
- **N** You have previously introduced RL at a token level, now (line 149) you discuss it on a response level.

## Significance

- **S** The method is relatively simple yet significant.
- **W5** The significance is limited by the evaluation on only a single domain and tasks on which majority voting is appropriate.


## Originality

- **S** The work is original to the best of my knowledge. I am not perfectly familiar with all related work, however.
- **S** Particularly the difficulty filtering for self-instruct, the majority-vote based rewards, and the online setting, seem novel.
- **N** You may want to discuss https://www.arxiv.org/pdf/2506.06395 as concurrent work.

---

> ### Author Rebuttal · Authors · 2025-07-31
>
> Dear Reviewer dA2g,
>
> Thanks for your professional review! We are glad you recognize the originality and significance of our work, including (1) the online instruction filter and (2) the majority-vote-based reward method. We also sincerely appreciate your positive feedback on our methodology, experimental evaluation, and the overall structure of the paper
>
> We apologize for any inconvenience caused by our multiple comments in response to your feedback. Due to the number of issues, we address W1–6, M1, M5, and N3-4 here, and will respond to the remaining Minor and N comments at the start of the discussion phase. Thank you for your understanding.
>
> **[W1\&W5] Supplement experiments in other domain, and discuss in hard-to-verify scenarios**
>
> Thanks for your suggestion.
>
> (1) As suggested, we have additionally conducted experiments in the medical domain. Specifically, we used 500 randomly selected instructions from the MedQA [1] training set as seed data for our method, and compared it with supervised RL trained on the full 10.2k dataset (RL-GT). We evaluated both models on the medical benchmark (MedQA, PubMedQA [2], and NephSAP [3]). The results in Table R3 show that our SeRL can achieve performance on par with RL-GT baseline, demonstrating its effectiveness.
>
> (2) Our work focuses on combining self-instruction and self-rewarding methods to incentivize the reasoning abilities of LLMs in data-scarce settings. Our current experiments are conducted on tasks with deterministic final answers. However, we have observed that some self-consistency methods [4] also perform well on general tasks that are hard to verify, which supports the potential of applying our majority voting approach to such scenarios. We appreciate your suggestion and will explore this direction in future work.
>
> Table R1: Pass@1 performance comparison between our method and other baseline methods on medical benchmark.
>
> |Models|Methods|MedQA|PubMedQA|NephSAP|
> |-|-|-|-|-|
> |LLaMA3.2-3B-Instruct|Initial|53.4|54.4|24.2|
> ||RL-GT|**57.2**|55.0|26.1|
> ||SeRL (Ours)|56.9|**55.3**|**27.5**|
> |Qwen2.5-7B-Instruct|Initial|58.2|32.1|29.5|
> ||RL-GT|**59.4**|33.1|**31.2**|
> ||SeRL (Ours)|59.1|**34.8**|30.9|
>
> ---
>
> **[W2] Supplement SFT experiments.**
>
> Thanks for your suggestion. We conducted this experiment and compared SFT using the sampled truly correct responses of the model. As shown in Table R2, we observe a consistent drop in performance. This may be due to the limited quality of self-sampled responses or the model's existing proficiency on those samples, resulting in minimal gains from further fine-tuning. To validate our findings, we additionally conduct SFT using data distilled from Qwen2.5-Math-7B-Instruct, and as we hypothesized, fine-tuning with higher-quality data is indeed necessary to enhance the capabilities of the model.
>
> Table R2: Pass@1 performance of LLaMA-3.2-3B-Instruct fine-tuned with different datasets.
>
> |Methods|MATH-500|MATH-Hard|ASDiv|College Math|TabMWP|
> |-|-|-|-|-|-|
> |Initial|47.6|22.5|84.6|35.2|46.4|
> |SFT on Majority responses|44.6|20.2|83.1|30.5|48.2|
> |SFT on Correct responses|44.6|20.7|83.9|33.0|50.6|
> |SFT on Distilled responses|49.0|23.6|87.1|36.4|66.1|
>
> ---
>
> **[W3] How does your chain-of-thought approach work?**
>
> Thanks for the constructive question. Following prior work [5-6], we prompt the model with “think step by step and give the final answer”, which encourages the model to first produce a chain-of-thought (CoT) followed by the final answer.
>
> ---
>
> **[W4] Explanation of rule-based and model-based rewards.**
>
> Sorry for the confusion. Rule-based methods [7-8] determine correctness by extracting the answer from the response using regular expressions and comparing it against the ground-truth answer to assign a reward. In contrast, model-based methods [9] rely on manually designed principles or scoring rubrics, which the model uses to evaluate the response and assign a reward.
>
> ---
>
> **[W6] Limitation discussion:**
>
> - **(1) Applicability of majority voting**
> - **(2) Single domain experiment.**
> - **(3) Mode collapse risk.**
> - **(4) Single seed evaluation.**
> - **(5) Reinforcement bias in ambiguous domains.**
>
> Thanks for your valuable suggestions.
>
> - (1) In scenarios where there is no deterministic final answer such as writing tasks, our majority voting strategy may not be applicable. However, for most knowledge-based scenarios, even when a numeric answer is not required as in mathematical problem solving, the question can often be reformulated into a multiple-choice format, making our majority voting method applicable.
> - (2) As suggested, we have additionally conducted experiments in the medical domain. (See Table R1.)
> - (3) We acknowledge that majority-vote-based self-rewarding carries the risk of collapsing to a single response. However, our online instruction filtering module helps mitigate this issue. In particular, the difficulty filtering strategy removes instructions for which the model is uncertain about the correct answer. As shown in Section 4.3 of the original manuscript, this strategy helps reduce potential bias during training.
> - (4) As suggested, we have additionally conducted experiments with different random seeds, as shown in Table R3. Due to time constraints, we will directly include other results in the revision.
> - (5) As stated in the related works [9, 10], estimation bias is inevitable in self-rewarding settings. Our majority-voting method assigns rewards based on the consistency among multiple sampled responses, offering greater stability than scoring individual responses. We will include a more detailed discussion of this issue in the revision.
>
> Table R3: Pass@1 results over multiple iterations on mathematical benchmarks.
>
> |Models|Methods|MATH-500|MATH-Hard|ASDiv|College Math|TabMWP|
> |-|-|-|-|-|-|-|
> |LLaMA3.2-3B-Instruct|Initial|48.1$\pm$1.1|22.4$\pm$0.5|85.6$\pm$0.7|34.9$\pm$0.6|46.7$\pm$0.9|
> ||RL-GT| 50.1$\pm$1.2|23.9$\pm$0.5|89.4$\pm$0.6|36.3$\pm$0.4|70.2$\pm$0.8|
> ||SeRL (iter1)|48.3$\pm$1.5|22.8$\pm$0.7|87.2$\pm$0.4|36.8$\pm$0.6|68.1$\pm$0.7|
> ||SeRL (iter2)|49.8$\pm$1.4|23.0$\pm$0.6|88.4$\pm$0.5|37.5$\pm$0.5|72.4$\pm$0.8|
> ||SeRL (iter3)|51.3$\pm$1.4|23.2$\pm$0.5|89.1$\pm$0.5|37.4$\pm$0.5|70.3$\pm$0.9|
> |Qwen2.5-7B-Instruct|Initial|74.4$\pm$1.2|47.8$\pm$0.9|93.4$\pm$0.4|54.3$\pm$1.1|75.7$\pm$1.0|
> ||RL-GT|74.9$\pm$1.1|50.8$\pm$0.8|94.6$\pm$0.4|55.4$\pm0.7$|73.6$\pm$0.8|
> ||SeRL (iter1)|74.4$\pm$1.1|50.6$\pm$0.8|94.4$\pm$0.3|54.8$\pm$0.4|77.8$\pm$0.6|
> ||SeRL (iter2)|75.6$\pm$1.1|49.4$\pm$0.9|94.5$\pm$0.3|55.0$\pm$0.4|79.5$\pm$1.1|
> ||SeRL (iter3)|75.2$\pm$1.2|50.7$\pm$0.8|94.7$\pm$0.3|55.0$\pm$0.5|79.3$\pm$0.8|
>
> ---
>
> **[M1\&M5\&N3] Explanation of Reinforce++, and computation of return and advantage.**
>
> Sorry for the confusion. Based on the return $G(s_t, y_t)$ defined in the original manuscript (Page 2, Line 77), we further clarify that, assuming a training batch contains $n_{\text{bs}}$ questions and each question is associated with $n_{\text{vote}}$ sampled responses, $G_{ij}(s_t, y_t)$ denotes the return at the $t$-th token of the $j$-th sampled response for the $i$-th question. And then, we compute the mean and standard deviation of the return across the batch as follows:
>
> $$
> \text{mean}\_t = \frac{1}{n_{\text{bs}} \cdot n_{\text{vote}}} \sum_{i=1}^{n_{\text{bs}}} \sum_{j=1}^{n_{\text{vote}}} G_{ij}(s_t, y_t), \quad
> \text{std}\_t = \sqrt{\frac{1}{n_{\text{bs}} \cdot n_{\text{vote}}} \sum_{i=1}^{n_{\text{bs}}} \sum_{j=1}^{n_{\text{vote}}} \left( G_{ij}(s_t, y_t) - \text{mean}\_t \right)^2}
> $$
>
> Then, the advantage is computed as:
>
> $$
> A_{ij}(s_t, y_t) = \frac{G_{ij}(s_t, y_t) - \text{mean}_t}{\text{std}_t}
> $$
>
> We would like to further clarify the computation of $G_{ij}(s_t, y_t)$ as follows:
>
> $$
> G_{ij}(s_t, y_t) = \sum_{k=t}^T R_{ij}(s_k, y_k) - KL_{ij}(k)
> $$
>
> where $R_{ij}(s_k, y_k)$ denotes the token-level reward at the $k$-th token of the $j$-th response to the $i$-th question and $KL_{ij}(k)$ denotes the token-level KL divergence at the $k$-th token between the current model and the initial model for the $j$-th response to the $i$-th question. A negative sign is applied to serve as a KL penalty.
>
> In practice, the reward is assigned at the response level, consistent with other related works [5-8]. Therefore, only the last token receives a reward, and the above equation (3) can be simplified as:
>
> $$
> G_{ij}(s_t, y_t) = R_{ij}(s_T, y_T) - \sum_{k=t}^T KL_{ij}(k) = R(x_i, y_j) - \sum_{k=t}^T KL_{ij}(k)
> $$
>
> Here, $R(x_i, \mathbf{y}_j)$ denotes the reward assigned to the $j$-th response of the $i$-th question.
>
> Note that in some algorithms, the KL penalty term $KL_{ij}(k)$ may be omitted. In such cases, if all responses receive the same reward, the resulting return would also be identical, leading to an advantage of zero. Even when the KL penalty is included, the advantage may still approximate zero. We will update the above explanations in the revision.
>
> ---
>
> **[N4] Reward granularity.**
>
> Sorry. The reward is assigned at the response level, consistent with other related works [5-8]. We will clarify and ensure consistency in the revision.
>
> ---
>
> **Reference**
>
> [1] Can Generalist Foundation Models Outcompete Special-Purpose Tuning? Case Study in Medicine. arXiv 2023
>
> [2] PubMedQA: A Dataset for Biomedical Research Question Answering. ACL 2019
>
> [3] Benchmarking Open-Source Large Language Models GPT-4 and Claude 2 on Multiple-Choice Questions in Nephrology. NEJM AI 2024
>
> [4] Universal Self-Consistency for Large Language Model Generation. arXiv 2023
>
> [5] Reinforcement Learning for Reasoning in Large Language Models with One Training Example. arXiv 2025
>
> [6] Reinforcing General Reasoning without Verifiers. arXiv 2025
>
> [7] DeepSeekMath: Pushing the Limits of Mathematical Reasoning in Open Language Models. arXiv 2024.
>
> [8] Iterative Reasoning Preference Optimization. arXiv 2024
>
> [9] Self-Rewarding Language Models. ICML 2024
>
> [10] Right Question is Already Half the Answer: Fully Unsupervised LLM Reasoning Incentivization. arXiv 2025

---

> ### Author Response · Authors · 2025-08-01
> **Additional Response (1/2)**
>
> **[M11] Comparison details with other self-instruction methods.**
>
> Thanks for the suggestion. These works focus on instruction generation. Zhao et al. [1] generate instructions with high temperature and I/O pairs with low temperature, using keyword and Gaussian-based length filtering. Galloy et al. [2] follow the prior work [3] and adapt the method to behavior-driven development. Kim et al. [4] raise the ROUGE-L threshold to reduce filtering and maintain intra-batch diversity during training. Gao et al. [5] use self-instruction with perplexity-based filtering and curriculum learning for tool-use training. Guo et al. [6] improve response reliability by using kNN retrieval for few-shot example selection.
>
> There are four main differences between our work and theirs:
>
> - (1) We generate only instructions to avoid noisy rewards from model-generated answers, use a low temperature of 0.6 to reduce off-topic instructions, and follow the same ROUGE-L threshold as Wang et al. [3] to maintain global instruction diversity.
> - (2) For the length filter, we adopt a simple and effective absolute token-length threshold to remove overly short or long instructions. This is because instructions shorter than the threshold typically lack sufficient information, while overly long instructions tend to contain redundant words or even include solutions.
> - (3) We design a difficulty filtering strategy inspired by active learning [7] to ensure that the model first learns from data of appropriate difficulty.
> - (4) Our work focuses on combining self-instruction and self-rewarding methods to incentivize the reasoning abilities of LLMs in data-scarce settings, aiming to achieve performance comparable to training on full high-quality data with verifiable rewards. In contrast, the above works only focus on instruction generation.
>
> ---
>
> **[M12] Compare all the self-rewarding methods mentioned in related work with the proposed method.**
>
> Thanks for your valuable feedback. These works primarily focus on the design of self-rewarding methods. Yuan et al. [8] design model-based reward methods by asking the model to score responses based on predefined principles. Bai et al. [9] similarly train a reward model based on predefined principles, enabling the model to assign rewards accordingly. Franken et al. [10] also relies on such principles, aiming to maximize the mutual information between the principles and responses, although it does not directly produce scalar rewards. Zhang et al. [11] propose a semantic-entropy-based method that assigns higher rewards to high-probability responses.
>
> There are two main differences between our work and theirs:
>
> - (1) We introduce a new majority-vote-based self-rewarding method. Since the method of Franken et al. does not output explicit rewards, we compare our approach against both model-based and entropy-based reward methods. As shown in Table R4, our method achieves higher reward accuracy.
> - (2) Our work focuses on combining self-instruction and self-rewarding methods to incentivize the reasoning abilities of LLMs in data-scarce settings, aiming to achieve performance comparable to training on full high-quality data with verifiable rewards. In contrast, the above works only focus on self-rewarding estimation.
>
> Table R4: Similarity between different self-rewards and the ground-truth rewards on MATH500 using LLaMA3.2-3B-Instruct. We assess the alignment between self-rewards and ground-truth rewards of all responses to MATH500 by treating them as vectors and computing similarity metrics; see Figure 3 of the original manuscript for details.
>
> | Methods | Cosine (↑) | MAE (↓) | MSE (↓) |
> | - | - | - | - |
> | Majority-vote-based reward (Ours) | **0.75** | **0.30** | **0.60** |
> | Model-based reward [1] | 0.17 | 0.89 | 1.2 |
> | Entropy-based reward [3] | 0.65 | 0.45 | 0.65 |
> | CAI reward [4] | 0.01 | 1.0 | 1.41 |
>
> ---
>
> **Reference**
>
>
> [1] Self-guide: Bette task-specific instruction following via self-synthetic finetuning. COLM 2024.
>
> [2] Selfbehave, generating a synthetic behaviour-driven development dataset using self-instruct. ICSTW 2025.
>
> [3] Self-Instruct: Aligning Language Models with Self-Generated Instructions. arXiv 2022.
>
> [4] Sedi-instruct: Enhancing alignment of language models through self-directed instruction generation. arXiv 2025.
>
> [5] Confucius: iterative tool learning from introspection feedback by easy-to-difficult curriculum. AAAI 2024.
>
> [6] Human instruction-free llm self-alignment with limited samples. arXiv 2024.
>
> [7] Strategic Data Ordering: Enhancing Large Language Model Performance through Curriculum Learning. arXiv 2024.
>
> [8] Self-Rewarding Language Models. ICML 2024.
>
> [9] Constitutional AI: Harmlessness from AI Feedback. arXiv 2022.
>
> [10] Self-supervised alignment with mutual information: Learning to follow principles without preference labels. NeurIPS 2024.
>
> [11] Right Question is Already Half the Answer: Fully Unsupervised LLM Reasoning Incentivization. arXiv 2025.

---

> ### Author Response · Authors · 2025-08-01
> **Additional Response (2/2)**
>
> **[M2] The background of self-instruction.**
>
> Thanks for your feedback. In data-scarce scenarios, the limited amount of data is insufficient to improve model performance through training. To address this, we adopt a self-instruction method, where seed data are used to prompt the model to generate more instructions in a few-shot manner. Moreover, to ensure that the generated instructions better align with the evolving capabilities of the model, we perform online instruction generation. Specifically, after each gradient update, we use the updated model to generate new instructions for the next step of gradient update. We will update the background in the revision.
>
> ---
>
> **[M3] Details on how few-shot is selected.**
>
> Thanks for the comment. In few-shot learning, the $n_{\text{shot}}$ examples are randomly sampled from both the seed data and the generated instructions. Specifically, one-fourth ($n_{\text{shot}}/4$) are drawn from the seed data, while the remaining three-fourths ($3n_{\text{shot}}/4$) from previously generated instructions. We will update this in the revision.
>
> ---
>
> **[M4] Details on filtering. For example, which words are filtered?**
>
> Sorry for the confusion. We filter out instructions containing specific keywords such as "image", "graph", "picture", "file", "map", "draw", "plot", or "write a program", as they refer to visual content or capabilities beyond the model's scope. In addition, instructions starting with punctuation or non-English characters are also excluded.
>
> ---
>
> **[M6] References on the vanishing gradients.**
>
> Thanks for pointing this out. The issue of vanishing gradients when using discrete or identical rewards has been discussed in prior work. Specifically, ReDit [1] (Sec. 3.1) notes that *...the gradient norm approaches zero (red dot), occurring when most examples in a GRPO batch yield identical binary rewards.* Similarly, GHPO [2] (Sec. 2.3) points out that *a zero advantage results in a vanishing policy gradient for that specific query.* We will include these references in the revision.
>
> ---
>
> **[M7] How does the number of gradient steps in your approach compare to the baselines?**
>
> Thanks for your question. To ensure a fair comparison with the baselines, we use the same amount of training data and performed an equal number of gradient steps. Specifically, we generate an equal number of instructions as in RL-GT, based on the limited seed data.
>
> ---
>
> **[M8] Why does RL-GT improve other but SeRL does not?**
>
> Thanks for your question. RL-GT is trained on the full MATH training set with verifiable rewards, and thus serves as the potential upper bound of our method. Its stronger performance on the *Social* and *Other* subsets of MMLU-Pro may be attributed to the broader distribution of the MATH training set.
>
> ---
>
> **[M9] Supplement the "with clipped" variant in Figure 2 for comparison.**
>
> Thanks for the suggestion. Since we are currently unable to upload images, we will include them in the revision. After applying the difficulty filter ("with clipped"), both the reward and accuracy increase steadily. We observe that as the reward approaches 0.53 and the accuracy, which is evaluated on the MATH-500 set after each gradient update, reaches 0.51, the upward trend gradually slows down.
>
> ---
>
> **[M10] Supplement more training plots like Figure 2 in addition to the final results, to see if performance saturates.**
>
> Thanks for the helpful suggestion. We will include more training plots in the revision.
>
> ---
>
> **[N1] It is a bit hard to understand what underline in the tables mean, especially as the meaning differs between the two tables.**
>
> Sorry for the confusion. The underlines are used to indicate the best-performing result of each model across all iterations within a given column. Since Table 1 of the original manuscript includes two models, each column may contain two underlined entries. We will make this clearer in the revision.
>
> ---
>
> **[N2] I think it is not entirely correct to call the observed behavior reward hacking, since the reward itself is modified. It seems like a sort of collapse.**
>
> Thanks for the suggestion. We will make the correction in the revision.
>
> ---
>
> **[N5] Discuss https://www.arxiv.org/pdf/2506.06395 as concurrent work.**
>
> Thanks for the additional information. RLSC [3] optimizes the model by directly maximizing its most probable response, while our majority voting method is based on multiple responses, leading to greater stability and more consistent outputs.
>
> ---
>
> **[Paper Formatting]**
>
> We appreciate your careful review and the detailed feedback on the writing format. We will address these issues in the revision.
>
> ---
>
> If you have further questions, please feel free to reach out.
>
> **Reference**
>
> [1] ReDit: Reward Dithering for Improved LLM Policy Optimization. arXiv 2025.
>
> [2] GHPO: Adaptive Guidance for Stable and Efficient LLM Reinforcement Learning. arXiv 2025.
>
> [3] Confidence Is All You Need: Few-Shot RL Fine-Tuning of Language Models. arXiv 2025.

---

> > ### Comment · Reviewer_dA2g · 2025-08-04
> >
> > Thank you for your detailed response. It addresses most of my concerns. In your revision, it would be helpful to clarify what exactly you mean by the "distilled response" in your additional experiment. Additionally, you should further clarify the setup of your chain of thought approach, e.g., list the exact prompt.
> >
> > Independent of these further clarifications, which I believe are minor, I will raise my score to recommend acceptance (5). I believe your method is a valuable contribution to data-scarce settings.

---

> ### Author Response · Authors · 2025-08-04
> **We sincerely appreciate the decision of the reviewer to raise the score.**
>
> We sincerely appreciate your positive feedback and are grateful for raising the score. Your insightful comments have been instrumental in helping us improve the quality of our work.
>
> Thanks for your questions. (1) By "distilled response," we refer to the responses generated by the Qwen2.5-Math-7B-Instruct model. We use these distilled responses to fine-tune LLaMA3.2-3B-Instruct, and as shown in Table R2, the model's performance improves compared to using its own self-sampled responses. This supports our statement in the response of W2: “the limited quality of self-sampled responses or the model's existing proficiency on those samples, resulting in minimal gains from further fine-tuning.”
>
> (2) Following prior work [1-4], the exact prompt is:
>
> ```
> {{Question}}
> Let's think step by step and output the final answer within \\boxed{}.
> ```
>
> Thank you again, and we wish you a wonderful day!
>
> **Reference**
>
> [1] Reinforcement Learning for Reasoning in Large Language Models with One Training Example. arXiv 2025.
>
> [2] Reinforcing General Reasoning without Verifiers. arXiv 2025.
>
> [3] DeepSeekMath: Pushing the Limits of Mathematical Reasoning in Open Language Models. arXiv 2024.
>
> [4] Tulu 3: Pushing Frontiers in Open Language Model Post-Training. arXiv 2024.

---

### Official Review · Reviewer_fTmJ · 2025-07-03

**Clarity:** 3
**Significance:** 3
**Originality:** 2
**Rating:** 4
**Confidence:** 4

**Summary:**

This paper introduces SeRL (Self-play Reinforcement Learning), a framework for training Large Language Models with limited initial data by leveraging self-instruction and self-rewarding modules. Self-instruction generates high-quality instructions based on available data, while self-rewarding estimates response rewards via majority voting, eliminating the need for external annotations. SeRL achieves performance comparable to models trained with extensive, high-quality data and outperforms other self-play methods across various reasoning benchmarks.

**Questions:**

1. In the filtering strategy introduced in Section 3.1, how is filtering based on average accuracy implemented? My main confusion is how the ground truth answer is obtained. I noticed that in Table C.1, there is no requirement to generate the gt simultaneously when generating the question. If gt is obtained through subsequent voting, what methods are there to ensure its reliability?

**Ethical Concerns:**

["NO or VERY MINOR ethics concerns only"]

**Final Justification:**

I confirm that the authors have adequately addressed most of the initial concerns and questions.

**Quality:**

3

**Strengths And Weaknesses:**

**Strengths**

- The idea of enabling model self-play through self-instruction and self-rewarding is interesting.
- The authors demonstrate the effectiveness of SeRL across multiple benchmarks, achieving results comparable to those trained with full high-quality data and verifiable rewards.

**Weaknesses**

- In Table 1, the performance improvement of the model slows down as the number of iterations increases. In Table 4, the performance even decreases during the iterations on some benchmarks and algorithms. Does this imply that the SeRL framework cannot support more rounds of self-play iterations, which is key to continual self-improvement? It is suggested that the authors supplement the results with more iterations (e.g., iter4, 5, 6, etc.).
- In this paper, the authors mainly conduct evaluations on math-related benchmarks. However, one of the claimed contributions is to address the problem of high-quality data and verifiable supervision scarcity in specialized domains such as clinical diagnostics and aerospace engineering. It is suggested that the authors supplement evaluations in these scenarios to illustrate the generalizability of the method and support the claimed contributions.
- The method of Self-instruction performs data augmentation based on seed data, which leads to the issue that the distribution of generated problems is strongly correlated with the seed data. Will this affect the model's generalization ability?
- The self-rewarding mechanism relies heavily on the reasoning capabilities of the policy model itself. For challenging problems (e.g., AIME-level questions), the reliability of the reward signal may be compromised since it may not obtain a correct answer through voting. However, this is a common issue with current Self-Rewarding Methods, so it is not my core concern.

---

> ### Author Rebuttal · Authors · 2025-07-31
>
> Dear Reviewer fTmJ,
>
> Thanks for your professional review! We are glad you recognize our self-play approach and its key contribution in achieving strong performance with limited data.
>
> **[W1] Results of more rounds and issues of continual self-improvement.**
>
> Thanks for the comment. (1) To ensure a fair comparison, we follow the same three-round training setup as self-rewarding baseline [1]. As suggested, we have additionally reported results for 4 to 6 rounds in Table R1, where we observe that the performance of the model gradually converges. As discussed in Section 6 of the original manuscript, we believe this convergence behavior is largely due to the limited capacity of the underlying foundation model. This observation aligns with findings from EMPO [2], which suggests that pre-trained language models already possess strong reasoning capabilities. In this context, RL post-training may primarily help activate latent reasoning patterns learned during pretraining, rather than introduce new ones.
>
> (2) It is worth emphasizing that our goal is not to achieve unlimited iterative improvement through self-play, which is currently an unrealistic expectation and an unsolved problem across the field. As stated in Introduction of the original manuscript, our work focuses on combining self-instruction and self-rewarding methods to incentivize the reasoning abilities of LLMs in data-scarce settings, aiming to achieve performance comparable to training on full high-quality data with verifiable rewards.
>
> Table R1: Multi-round performance on the math-specific benchmark.
> | Models | Methods | MATH-500 | MATH-Hard | ASDiv | College Math | TabMWP |
> | - | - | - | - | - | - | - |
> | LLaMA3.2-3B-Instruct | SeRL (iter1) | 48.6 | 23.0 | 87.5 | 36.7 | 68.4 |
> | | SeRL (iter2) | 50.4 | 23.6 | 88.9 | **38.2** | **72.3** |
> | | SeRL (iter3) | 52.6 | 23.7 | 89.0 | 37.7 | 70.6 |
> | | SeRL (iter4) | **52.7** | 24.1 | 88.8 | 38.0 | 71.1 |
> | | SeRL (iter5) | 52.1 | 23.9 | 88.9 | 37.9 | 71.5 |
> | | SeRL (iter6) | 52.0 | **24.2** | **89.1** | 37.9 | 70.9 |
> | Qwen2.5-7B-Instruct | SeRL (iter1) | 74.2 | 50.0 | 94.2 | 54.3 | 77.0 |
> | | SeRL (iter2) | 74.8 | 49.7 | **94.7** | 55.1 | 80.1 |
> | | SeRL (iter3) | 75.8 | 50.4 | 94.4 | 55.1 | 79.4 |
> | | SeRL (iter4) | 75.4 | **50.6**  | 94.4 | 55.2 | 79.7 |
> | | SeRL (iter5) | 76.0 | 50.1 | 94.6 | **55.3** | 80.2 |
> | | SeRL (iter6) | **76.1** | 49.8 | 94.6 | **55.3** | **80.4** |
>
> ---
>
> **[W2] In this paper, the authors mainly conduct evaluations on math-related benchmarks. However, one of the claimed contributions is to address the problem of high-quality data and verifiable supervision scarcity in specialized domains such as clinical diagnostics and aerospace engineering. It is suggested that the authors supplement evaluations in these scenarios to illustrate the generalizability of the method and support the claimed contributions.**
>
> Thanks for your suggestion. As suggested, we have additionally conducted experiments in the medical domain. Specifically, we used 500 randomly selected instructions from the MedQA [4] training set as seed data for our method, and compared it with supervised RL trained on the full 10.2k dataset (RL-GT). We evaluated both models on the medical benchmark (MedQA, PubMedQA [5], and NephSAP [6]). The results in Table R3 show that our SeRL can achieve performance on par with RL-GT baseline, demonstrating its effectiveness.
>
> Table R3: Pass@1 performance comparison between our method and other baselines on the medical benchmark.
>
> | Models | Methods | MedQA | PubMedQA | NephSAP |
> | - | - | - | - | - |
> | LLaMA3.2-3B-Instruct | Initial | 53.4 | 54.4 | 24.2 |
> | | RL-GT | **57.2** | 55.0 | 26.1 |
> | | SeRL (Ours) | 56.9 | **55.3** | **27.5** |
> | Qwen2.5-7B-Instruct | Initial | 58.2 | 32.1 | 29.5 |
> | | RL-GT | **59.4** | 33.1 | **31.2** |
> | | SeRL (Ours) | 59.1  | **34.8** | 30.9 |
>
> ---
>
> **[W3] The method of Self-instruction performs data augmentation based on seed data, which leads to the issue that the distribution of generated problems is strongly correlated with the seed data. Will this affect the model's generalization ability?**
>
> Thanks for the insightful comment.
>
> - (1) It is worth noting that, in data-scarce scenarios, the limited seed data serves as the only available source of prior information about the target domain. Thus, our goal is to reconstruct the underlying distribution of this seed data, thereby mitigating the effects of data scarcity and enabling more effective model training.
> - (2) To ensure generalization, we further introduce an online filtering strategy that maintains data quality while promoting diversity in the generated samples. We have provided a comprehensive analysis of the generated data in terms of quality, difficulty, and diversity, as detailed in Appendix E of the original supplementary material. These results demonstrate the reliability of our self-instruction method for data generation.
> - (3) From an experimental perspective, training on data generated from seed data in the mathematical domain not only enhances the model's mathematical reasoning capabilities (see Table 1 of the original manuscript). It also improves its performance on general reasoning tasks (see Table 2 of the original manuscript), demonstrating strong generalization ability.
>
> ---
>
> **[W4] The self-rewarding mechanism relies heavily on the reasoning capabilities of the policy model itself. For challenging problems (e.g., AIME-level questions), the reliability of the reward signal may be compromised since it may not obtain a correct answer through voting. However, this is a common issue with current Self-Rewarding Methods, so it is not my core concern.**
>
> Thanks for the feedback. We agree that the reliability of the reward signal in self-rewarding methods can be compromised for challenging problems, as also discussed in prior work [1-3]. To mitigate this issue, we have designed a difficulty filtering strategy that removes instructions for which the model shows uncertainty in generating consistent answers. This acts as a form of curriculum learning, where the model starts training on simpler problems and gradually moves toward more complex ones. As a result, the estimation bias induced by unreliable self-rewards is reduced during training. While this strategy does not completely resolve the issue, we believe it is a promising direction and worth further exploration in future work. We sincerely appreciate the reviewer’s insightful comment.
>
> ---
>
> **[Q1] In the filtering strategy introduced in Section 3.1, how is filtering based on average accuracy implemented? My main confusion is how the ground truth answer is obtained. I noticed that in Table C.1, there is no requirement to generate the gt simultaneously when generating the question. If gt is obtained through subsequent voting, what methods are there to ensure its reliability?**
>
> Sorry for the confusion. (1) Yes. Our self-instruction module only generates questions without the corresponding answers. As stated in the original manuscript (Page 2, Lines 46–47), this is because directly treating model-generated answers as ground truth may introduce noise and inconsistency, thereby leading to unreliable reward signals for RL.
>
> (2) As described in the original manuscript (Page 4, Lines 130-135), the majority answer among different sampled responses is treated as the ground truth. For a given question, let $n_{\text{vote}}$ be the total number of responses and $n_{\text{maj}}$ the number of responses agreeing with the majority answer. We define the average accuracy as  $n_{\text{maj}} / n_{\text{vote}}$. Finally, we filter the instructions whose average accuracy falls outside the difficulty interval.
>
> (3) As stated in the related works [1-3], estimation bias is inevitable in self-rewarding settings. However, we employ two strategies to mitigate this bias:
>
> - Our majority-voting method assigns rewards based on the consistency among multiple sampled responses, offering greater stability than scoring individual responses. To demonstrate this, we have additionally conducted experiments with more self-rewarding baselines. The results in Table R4 show that our majority-vote-based reward method achieves higher accuracy, indicating lower bias compared to other self-rewarding methods.
>
> - In addition, our difficulty filtering strategy removes instructions for which the model is uncertain about the correct answer. As shown in Section 4.3 of the original manuscript, this strategy also further reduces potential bias during training.
>
> Finally, our proposed SeRL can achieve performance on par with those obtained by verifiable ground-truth rewards (RL-GT baseline), as shown in Table 1 of the original manuscript.
>
>
> Table R4: Similarity between different self-rewards and the ground-truth rewards on MATH500 using LLaMA3.2-3B-Instruct. We assess the alignment between self-rewards and ground-truth rewards of all responses to MATH500 by treating them as vectors and computing similarity metrics; see Figure 3 of the original manuscript for details.
>
> | Methods | Cosine (↑) | MAE (↓) | MSE (↓) |
> | - | - | - | - |
> | Majority-vote-based reward (Ours) | **0.75** | **0.30** | **0.60** |
> | Model-based reward [1] | 0.17 | 0.89 | 1.2 |
> | Entropy-based reward [3] | 0.65 | 0.45 | 0.65 |
> | CAI reward [4] | 0.01 | 1.0 | 1.41 |
>
> ---
>
> If you have further questions, feel free to let us know.
>
> **References**
>
> [1] Self-Rewarding Language Models. ICML 2024
>
> [2] Right Question is Already Half the Answer: Fully Unsupervised LLM Reasoning Incentivization. Arxiv 2025
>
> [3] TTRL: Test-Time Reinforcement Learning. arXiv 2025
>
> [4] Can Generalist Foundation Models Outcompete Special-Purpose Tuning? Case Study in Medicine. arXiv 2023
>
> [5] PubMedQA: A Dataset for Biomedical Research Question Answering. ACL 2019
>
> [6] Benchmarking Open-Source Large Language Models, GPT-4 and Claude 2 on Multiple-Choice Questions in Nephrology. NEJM AI 2024
>
> [7] Constitutional AI: Harmlessness from AI Feedback. arXiv 2022

---

> > ### Comment · Reviewer_fTmJ · 2025-08-04
> >
> > Thank you for the detailed responses. I confirm that the authors have adequately addressed most of the initial concerns and questions. I will raise my score.

---

> > > ### Author Response · Authors · 2025-08-04
> > > **We sincerely appreciate the decision of the reviewer to raise the score.**
> > >
> > > We are glad that the reviewer appreciates our attempt, and sincerely thank you for raising the score! Your constructive comments have greatly helped our paper to improve. Have a nice day :)

---

### Official Review · Reviewer_QSrT · 2025-07-04

**Clarity:** 3
**Significance:** 2
**Originality:** 2
**Rating:** 4
**Confidence:** 3

**Summary:**

This paper introduces SeRL, a framework for aligning large language models (LLMs) through a fully automated, self-bootstrapping RL process. SERL eliminates the need for any human-labeled preference data or reward models. The experiment results demonstrate that models trained with SeRL show strong performance, outperforming those trained with SR-DPO, I-RPO.

**Questions:**

1. How does SeRL compare with Constitutional AI in terms of resource use and stability?

2. What is the minimum evaluator quality required for SERL to work?

**Ethical Concerns:**

["NO or VERY MINOR ethics concerns only"]

**Final Justification:**

The response has addressed most of my concerns.

**Limitations:**

More discussion of failure modes and self-reward bias would be helpful.

**Quality:**

2

**Strengths And Weaknesses:**

Strengths
1. The paper is well written and easy to follow.

2. SERL provides a practical and robust framework that can significantly reduce the training cost.

Weaknesses

1. If I understand correctly, the self-reward strategy heavily relies on existing models, and there is limited discussion on how to choose the best self-reward models, and if the reward evaluator is miscalibrated or biased, it could result in suboptimal behaviors.

2. While comparisons are made to baselines, SeRL has conceptual overlap with Constitutional AI [1], but this is only briefly mentioned. A deeper empirical or theoretical comparison could enhance clarity.
[1] Bai, Yuntao, et al. "Constitutional ai: Harmlessness from ai feedback." arXiv preprint arXiv:2212.08073 (2022).

3. Some related works are missing, e.g., [2] and [3]

[2] Xu, Can, et al. "Wizardlm: Empowering large language models to follow complex instructions." arXiv preprint arXiv:2304.12244 (2023).

[3] Gu, Shangding, Alois Knoll, and Ming Jin. "TeaMs-RL: Teaching LLMs to Generate Better Instruction Datasets via Reinforcement Learning." arXiv preprint arXiv:2403.08694 (2024).

---

> ### Author Rebuttal · Authors · 2025-07-31
>
> Dear Reviewer QDrT,
>
> Thanks for your professional review! We are glad that you recognize the practicality and robustness of our method.
>
> **[W1] If I understand correctly, the self-reward strategy heavily relies on existing models, and there is limited discussion on how to choose the best self-reward models, and if the reward evaluator is miscalibrated or biased, it could result in suboptimal behaviors.**
>
> Thanks. (1) We follow the self-rewarding setting as prior works [1-3], where the *language model* under training also performs self-rewarding. Specifically, our self-rewarding module assigns rewards based on majority voting among multiple sampled responses, rather than directly scoring each response using the *language model*. This design naturally fits within the RL framework, where responses are sampled online during training. As a result, only the current version of the model is involved in response generation and reward computation, and no earlier model checkpoints are used. Therefore, the question of selecting the best self-reward model does not arise in this context.
>
> (2) As stated in the related works [1-3], estimation bias is inevitable in self-rewarding settings. However, we employ two strategies to mitigate this bias:
>
> - Our majority-voting method assigns rewards based on the consistency among multiple sampled responses, offering greater stability than scoring individual responses. To demonstrate this, we have additionally conducted experiments with more self-rewarding baselines. The results in Table R1 show that our majority-vote-based reward method achieves higher accuracy, indicating lower bias compared to other self-rewarding methods.
>
> - In addition, our difficulty filtering strategy removes instructions for which the model is uncertain about the correct answer. As shown in Section 4.3 of the original manuscript, this strategy also further reduces potential bias during training.
>
> Finally, our proposed SeRL can achieve performance on par with those obtained by verifiable ground-truth rewards (RL-GT baseline), as shown in Table 1 of the original manuscript.
>
> Table R1: Similarity between different self-rewards and the ground-truth rewards on MATH500 using LLaMA3.2-3B-Instruct. We assess the alignment between self-rewards and ground-truth rewards of all responses to MATH500 by treating them as vectors and computing similarity metrics; see Figure 3 of the original manuscript for details.
>
> | Methods | Cosine (↑) | MAE (↓) | MSE (↓) |
> | - | - | - | - |
> | Majority-vote-based reward (Ours) | **0.75** | **0.30** | **0.60** |
> | Model-based reward [1] | 0.17 | 0.89 | 1.2 |
> | Entropy-based reward [3] | 0.65 | 0.45 | 0.65 |
> | CAI reward [4] | 0.01 | 1.0 | 1.41 |
>
> ---
>
> **[W2\&Q1] Compare with Constitutional AI.**
>
> Thanks for the suggestion. Constitutional AI (CAI) [4] uses predefined principles for self-critique and self-revision, requiring substantial data for both SFT and RL, along with training a reward model prior to RL. In contrast, our method leverages self-instruction for data augmentation, making it effective even in data-scarce scenarios, and performs self-rewarding RL directly without the need for SFT warm-up or a trained reward model, resulting in lower resource consumption.
>
> As suggested, we have additionally conducted experiments using CAI. *CAI-SL* refers to the model after SFT warm-up in the CAI method, and *CAI-RL* denotes the model obtained by further applying RL training on top of *CAI-SL*. *CAI-RL (direct)* refers to the setting where RL is directly performed on the initial model using the reward model trained in the CAI method. The results in Table R2 show that our SeRL yields promising results, while the CAI method leads to a decline in model performance.
>
> - The performance drop of *CAI-SL* can be attributed to the fact that self-critique and self-revision rely more heavily on the model's own capability to correctly revise and improve its responses. To support this point, we have additionally conducted experiments to compare the change in accuracy before and after revision. As shown in Table R3, we observed that many initially correct responses were revised into incorrect ones.
>
> - The performance degradation of *CAI-RL* and *CAI-RL (direct)* stems from the low accuracy of the reward model trained using CAI. As shown in Table R1, this reward model achieves the lowest accuracy among all the self-rewarding methods. This poor performance is mainly due to how CAI generates preference pairs, relying on the model to make judgments based on predefined principles. When the base model is weak, the resulting preference data can be noisy, leading to a weak reward model and reduced RL training effectiveness.
>
> Additionally, the detailed comparison of resource consumption is as follows:
>
> - Training a single RL model using CAI takes about 24.6 8×A6000 GPU hours, which includes 2.8 hours for generating 7.5K training samples for SFT, 3.0 hours for SFT warmup, 2.4 hours for generating 7.5K training samples for the reward model, 2.8 hours for reward model training, and 13.6 hours for RL training. In contrast, our method requires only 11.2 8×A6000 GPU hours to reach the same number of training steps, demonstrating significantly higher training efficiency.
>
> - It is noted that our method use only 500 instructions as seed data, whereas CAI requires 7500 instructions, highlighting the data efficiency of our method.
>
> Table R2: Pass@1 performance comparison between CAI and SeRL using the LLaMA-3.2-3B-Instruct model.
>
> | Methods | MATH-500 | MATH-Hard | ASDiv | College Math | TabMWP |
> | - | - | - | - | - | - |
> | Initial | 47.6 | 22.5 | 84.6 | 35.2 | 46.4  |
> | CAI-SL | 30.0 | 10.9 | 67.1 | 21.9 | 38.5 |
> | CAI-RL | 26.4 | 9.9 | 64.2 | 19.2 | 31.8 |
> | CAI-RL (direct) | 42.2 | 15.6 | 83.5 | 32.5 | 63.5 |
> | SeRL (ours) | **48.6** | **23.0** | **87.5** | **36.7** | **68.4** |
>
> Table R3: Pass@1 accuracy on MATH-500 using LLaMA-3.2-3B-Instruct before (*Accuracy@t₁*) and after (*Accuracy@t₂*) applying self-critique and self-revision in CAI.
>  $\Delta(t_1, t_2)$ decnotes overall accuracy change; $\Delta^{i \rightarrow c}(\text{t}_1, \text{t}_2)$ denotes the proportion of responses corrected from incorrect to correct; $\Delta^{c \rightarrow i}(\text{t}_1, \text{t}_2)$ denotes the proportion of responses changed from correct to incorrect after revision.
>
> | Models | Accuracy@t1 | Accuracy@t2 | $\Delta(\text{t}_1, \text{t}_2)$ | $\Delta^{i\rightarrow c}(\text{t}_1, \text{t}_2)$ | $\Delta^{c\rightarrow i}(\text{t}_1, \text{t}_2)$ |
> | - | - | - | - | - | - |
> | LLaMA3.2-3B-Instruct | 47.6% | 41.6% | -6.0% | 10% | 4.0% |
>
> ---
>
> **[W3] Missing related works.**
>
> Thanks for the valuable feedback. These two works focus on instruction generation. WizardLM [5] improves instruction data by evolving it in terms of complexity, diversity, and quality. TeaMs-RL [6] builds upon WizardLM by defining a set of prompt-based instruction augmentation actions. It then trains an LLM via RL to apply these actions, resulting in an Instructor LLM capable of generating augmented instructions. There are two main differences between our work and theirs:
>
> - (1) In our self-instruction approach, we do not train a separate model or design additional instruction augmentation prompts. By leveraging few-shot generation along with our online filtering strategy, we observe that the model can generate diverse high-quality instructions, as detailed in Appendix E of the original supplementary material.
> - (2) Our work focuses on combining self-instruction and self-rewarding methods to incentivize the reasoning abilities of LLMs in data-scarce settings, aiming to achieve performance comparable to training on full high-quality data with verifiable rewards. In contrast, these two works only focus on instruction generation.
>
> These are two very interesting works, and we will discuss them further in the revision.
>
> ---
>
> **[Q2] Minimum evaluator quality required for SERL**
>
> Thanks for the insightful comments. (1) As noted in our W1 response, our method use majority voting over the model’s own samples to assign rewards, thus eliminating reliance on external evaluators. (2) As stated in our original manuscript (Page 9, Lines 321–325) and related self-rewarding works [1-3], RL post-training, whether supervised or unsupervised, may primarily help activate latent reasoning patterns learned during pretraining, rather than introduce new ones [3]. This typically improves Pass@1 by leveraging existing Pass@K capabilities. In our case, the self-rewarding mechanism is based on majority voting, where Maj@K reflects the effectiveness of the majority-voting reward. Therefore, our method can be effective when Maj@K exceeds Pass@1.
>
> ---
>
> **[L1] More discussion of failure modes and self-reward bias.**
>
> Thanks for your suggestions. (1) In scenarios where there is no deterministic final answer such as writing tasks, or where the correctness of the process is more important such as mathematical proofs, our majority voting strategy may not be applicable. However, for most knowledge-based scenarios, even when a numeric answer is not required as in mathematical problem solving, the question can often be reformulated into a multiple-choice format, making our majority voting approach still applicable. (2) We have additionally discussed the self-reward bias in the response to W1. As suggested, we will add these discussion in the revision.
>
> ---
>
> If you have further questions, please feel free to reach out.
>
> **References**
>
> [1] Self-Rewarding Language Models. ICML 2024.
>
> [2] TTRL: Test-Time Reinforcement Learning. arXiv 2025.
>
> [3] Right Question is Already Half the Answer: Fully Unsupervised LLM Reasoning Incentivization. arXiv 2025.
>
> [4] Constitutional AI: Harmlessness from AI Feedback. arXiv 2022.
>
> [5] WizardLM: Empowering large pre-trained language models to follow complex instructions. arXiv 2023.
>
> [6] TeaMs-RL: Teaching LLMs to Generate Better Instruction Datasets via Reinforcement Learning. arXiv 2024.

---

> ### Comment · Reviewer_QSrT · 2025-08-04
>
> Thank you for your detailed response, which has addressed most of my concerns. I updated my score accordingly.

---

> > ### Author Response · Authors · 2025-08-05
> > **We sincerely appreciate the decision of the reviewer to raise the score.**
> >
> > We are pleased that the reviewer recognized our efforts, and we sincerely thank you for raising the score. Your constructive comments have been very helpful in improving our paper. Wishing you a wonderful day!

---

### Author Response · Authors · 2025-08-09
**Response Summary for Submission 6968**

We deeply appreciate the time and effort the reviewers have dedicated to reviewing our submission. We are truly honored to have received feedback from four thoughtful reviewers. We have carefully considered and addressed all the constructive suggestions, and we are grateful for the opportunity to improve our work based on your valuable insights.

In this post:

- (1) We summarize positive comments from the reviewers.
- (2) We summarize the revisions of the manuscript.

### **(1) Positive Comments**

**Quality**

- `[dA2g]`: *"The submission is technically sound, all claims are supported by experimental evidence. The methods, metrics, and benchmarks used for evaluation are appropriate. The paper is complete without obvious missing parts."*
- `[MF7H]`: *"... The ablation study supports the claim of the effectiveness of the self-instruction and self-rewarding modules."*

**Clarity**

- `[QSrT]`: *"The paper is generally well-written, clear, and easy to follow."*
- `[dA2g]`: *"The paper is generally well-written and well-organized."*
- `[MF7H]`: *"The paper is well-structured in the presentation of the methodology and experiments."*

**Significance**

- `[dA2g]`: "*I believe your method is a valuable contribution to data-scarce settings.*"
- `[fTmJ]`: *"The authors demonstrate the effectiveness of SeRL across multiple benchmarks ..."*
- `[QSrT]`: *"SERL provides a practical and robust framework ..."*
- `[MF7H]`: *"In practice, it is hard to collect a large amount of data ... The proposed method provides a way to enlarge the dataset for the LLMs post-training."*

**Originality**

- `[fTmJ]`: *"The idea of enabling model self-play through self-instruction and self-rewarding is interesting."*
- `[dA2g]`: *"The work is original to the best of my knowledge." "Particularly the difficulty filtering for self-instruct, the majority-vote based rewards, and the online setting, seem novel."*
- `[MF7H]`: *"The self-rewarding with the vote-based mechanism is novel ..."*

### **(2) Revisions of the manuscript**

**Background**

1. `[dA2g]`: Additional details on the reinforce++, return, and advantage computation.
2. `[dA2g]`: Clarify that rewards are assigned at the response level.
3. `[dA2g]`: Additional details on the background of self-instruction.

**Methods**

1. `[QSrT]`: Clarify that the majority-vote-based reward method is different from external reward models based on neural networks or language.
2. `[QSrT, fTmJ, dA2g, MF7H]`: Explain how to reduce the bias introduced by self-rewarding and self-instruction.
3. `[fTmJ]`: Explain that self-instruction aims to generate data that aligns with the seed data distribution while introducing additional diversity.

**Experiments**

1. `[QSrT]`: Compare with Constitutional AI method.
2. `[fTmJ, MF7H]`: Additional results for 4-6 rounds of iteration.
3. `[fTmJ, dA2g]`: Additional results in the medical domain.
4. `[dA2g]`: Additional SFT baselines.
5. `[dA2g]`: Additional evaluation results under multiple seeds.
6. `[dA2g]`: Clarify experimental settings with the same training gradient steps as the baselines.
7. `[dA2g]`: Explain why RL-GT performs better on some benchmarks.
8. `[dA2g]`: Add the "with clipped" variant in Figure 2 for comparison.
9. `[MF7H]`: Additional ablation experiments on all data filter strategies
10. `[MF7H]`: Add analysis experiments of relationships between different data filter strategies.
11. `[MF7H]`: Explain why Reinforce++ is more robust.

**Discussion**

1. `[QSrT]`: Discuss the model requirements for majority voting.
2. `[QSrT, dA2g]`: Discuss scenarios where majority voting is applicable.
3. `[fTmJ]`: Discussion on challenging task.

**Appendix**

1. `[dA2g]`: Details on Chain of Thought (CoT).
2. `[dA2g]`: Explain rule-based and model-based methods.
3. `[dA2g]`: Conditions under which zero advantage appears.
4. `[dA2g]`: Details on how few-shot are selected.
5. `[fTmJ, dA2g, MF7H]`: Details on all filtering strategies.
6. `[dA2g]`: Additional training plots.
7. `[MF7H]`: Explain why use cosine similarity.

**Related Work**

1. `[QSrT, dA2g]`: Discuss works [1-3].
2. `[dA2g]`: Further details on the comparison of related works.

**Writing**

1. `[dA2g]`: Clarify the meaning of underlines in the tables.
2. `[dA2g]`: Add citations on vanishing gradients [4-5].
3. `[dA2g]`: Correct the term "reward hacking."
4. `[dA2g]`: Other grammar issues.

**References**

[1] WizardLM: Empowering large pre-trained language models to follow complex instructions.

[2] TeaMs-RL: Teaching LLMs to Generate Better Instruction Datasets via Reinforcement Learning.

[3] Confidence Is All You Need: Few-Shot RL Fine-Tuning of Language Models.

[4] ReDit: Reward Dithering for Improved LLM Policy Optimization.

[5] GHPO: Adaptive Guidance for Stable and Efficient LLM Reinforcement Learning.

---

Lastly, we would like to sincerely thank all the reviewers once again for your constructive feedback and valuable suggestions.

Best Regards,

Authors of Submission 6968.

---

### Decision · Program_Chairs · 2025-09-17

**Decision:**

Accept (poster)

**Comment:**

This paper proposes Self-play Reinforcement Learning (SeRL) to bootstrap LLM training with limited initial data. A self-instruction stage generates new instructions based on the available data at each training step with filtering, and a self-rewarding stage uses majority voting to estimate response rewards for the generated instructions. Finally, SeRL performs conventional RL based on the generated data.

The reviewers generally found the proposed method to be interesting and sound, though the individual components (e.g., self-rewarding, self-instruction) are not completely new. The empirical evaluations are solid and demonstrate the effectiveness of the method.

The authors provided extensive results in the rebuttal to address the initial concerns raised by the reviewers, and the reviewers were satisfied with the responses. After the rebuttal, all reviewers were positive about this paper.